



# Magnetospheric Response to Solar Wind Forcing: ULF Wave – Particle Interaction Perspective

**Qiugang ZONG** [1,2]*

[1] Institute of Space Physics and Applied Technology, Peking University, Beijing 100871, China

[2] Polar Research Institute of China, Shanghai 200136, China

* *Invited contribution by Qiugang Zong, recipient of the EGU Hannes Alfvén Medal 2020.*

Correspondence: Qiugang Zong (qgzong@pku.edu.cn)

**Abstract.**

Solar wind forcing, e.g. interplanetary shock and/or solar wind dynamic pressure pulses impact on the Earth's magnetosphere manifests many fundamental important space physics phenomena including producing electromagnetic waves, plasma heating and energetic particle acceleration. This paper summarizes our present understanding of the magnetospheric response to solar wind forcing in the aspects of radiation belt electrons, ring current ions and plasmaspheric

plasma physics based on in situ spacecraft measurements, ground-based magnetometer data, MHD and kinetic simulations.

Magnetosphere response to solar wind forcing, is not just a "one-kick" scenario. It is found that after the impact of solar wind forcing on the Earth's magnetosphere, plasma heating and energetic particle acceleration started nearly immediately and could last for a few hours. Even a small dynamic pressure change of interplanetary shock or solar wind pressure pulse can play a non-negligible role in magnetospheric physics. The impact leads to generate series kind of waves including

poloidal mode ultra-low frequency (ULF) waves. The fast acceleration of energetic electrons in the radiation belt and energetic ions in the ring current region response to the impact usually contains two contributing steps: (1) the initial adiabatic acceleration due to the magnetospheric compression; (2) followed by the wave-particle resonant acceleration dominated by global or localized poloidal ULF waves excited at various L-shells.

Generalized theory of drift and drift-bounce resonance with growth or decay localized ULF waves has been developed

to explain in situ spacecraft observations. The wave related observational features like distorted energy spectrum, boomerang and fishbone pitch angle distributions of radiation belt electrons, ring current ions and plasmaspheric plasma can be explained in the frame work of this generalized theory. It is worthy to point out here that poloidal ULF waves are much more efficient to accelerate and modulate electrons (fundamental mode) in the radiation belt and charged ions (second harmonic)

in the ring current region. The results presented in this paper can be widely used in solar wind interacting with other planets such as Mercury, Jupiter, Saturn, Uranus and Neptune, and other astrophysical objects with magnetic fields.

# 1 Introduction

"We have to learn again that science without contact with experiments is an enterprise which is likely to go completely astray into imaginary conjecture." (Evolution of the Solar System by Alfvén and Arrhenius, 1976).

## 1.1 Hannes Alfvén and China

The European Geosciences Union (EGU) has awarded the Hannes Alfvén Medal to me for the year 2020, I feel deeply honored and very happy to obtain so much recognition for my work, because Hannes Alfvén was one of the giants in space physics and astrophysics, and also one of my heroes.

As we all know, Alfvén received the 1970 Nobel Prize in physics for his work in magnetohydrodynamics (MHD) and plasma physics. While few people know that Hannes Alfvén can speak some Chinese (Fig. 1) besides Swedish and English as indicated by Wikipedia. In fact, he has visited China twice, his first visit was invited by Prof. Jeoujang Jaw who is the founder of Chinese space program [Zhang &Yin, 2018]. During his in total of 50 days' visiting in China, Hannes Alfvén has given a number of lectures and promoted China's space physics. Also, at the early 1990s, the first text book I took to learn space physics-- Cosmical Electrodynamics (the second edition, 1963) is in fact the gift from Hannes Alfvén during his first China visiting. Alfvén's Cosmical Electrodynamics contains the main fundamentals of space plasma physics.


Annales Geophysicae Open Access
Discussions
EGU

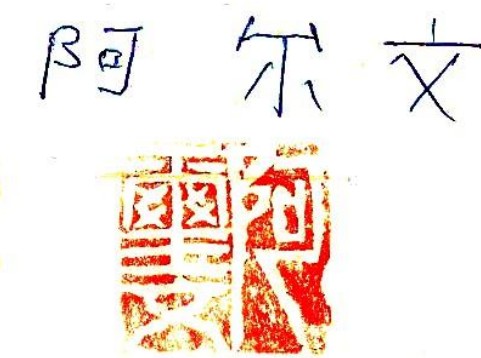

**Figure 1: Hannes Alfvén's hand writing signature in Chinese (top), and his name seal in traditional Chinese character**

**(**Credits: Xiaodong YIN)

As a student majored in space science, I first studied Alfvén's eminent works - the motion of charged particles (Alfvén and Fälthammar, 1963) and 'Existence of Electromagnetic–Hydrodynamic Waves' (Alfvén, 1942). The latter one is now named after him as Alfvén waves. 78 years after the publication of the paper 'Existence of Electromagnetic–Hydrodynamic Waves', Alfvén waves have "propagated" to plenty of regimes of cosmic plasmas. Now, it is understood that

Alfvén waves are ubiquitous and of fundamental importance in plasma physics, space physics & astrophysics, they occur in planetary magnetospheres, in laboratory plasma, in stellar coronas and winds, and many other astrophysical plasmas in the universe.

## 1.2 ULF waves and Solar wind forcing

Ultra-low frequency (ULF) waves are electromagnetic waves originated in the Earth's magnetosphere with frequency range from about 1 mHz to 10 Hz. Usually, ULF waves containing larger power are the higher frequency ones, the intensity of the





wave power in general has inverse relation with respect to its frequency [e.g. Lanzerotti and Southwood, 1979, Zong et al, 2017].

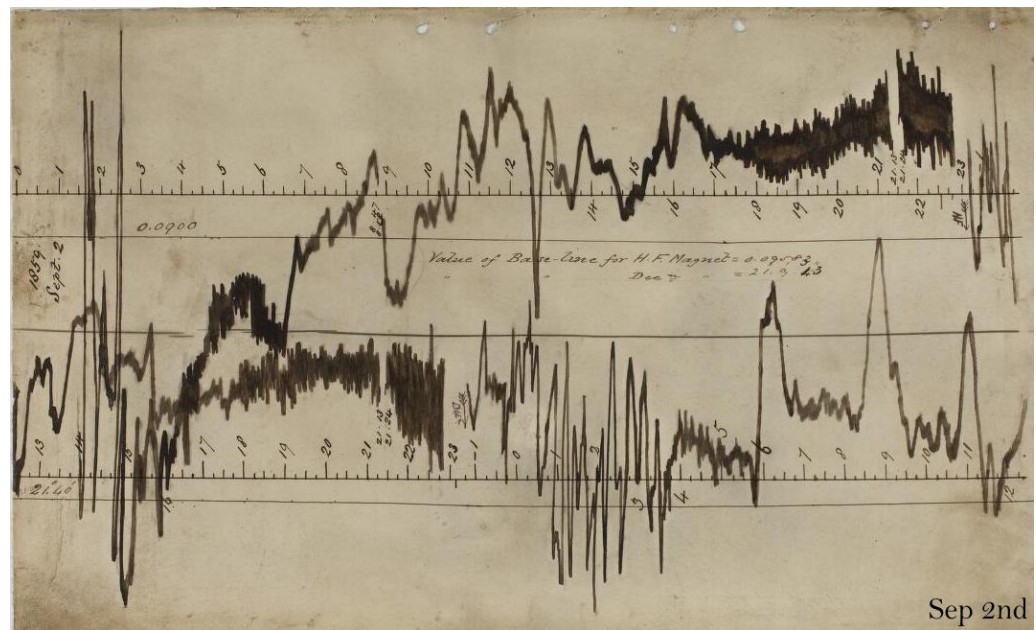

Figure 2: The 'Carrington Event' of September 2th, 1859 recorded at Greenwich Observatory, London (51.4769° N, 0.0005° W). https://geomag.bgs.ac.uk/education/carrington.html Greenwich Observatory magnetometer traces (horizontal force (H)

on top and declination (D) on the bottom; the two traces are offset by 12 h) during the time of the solar flare on 1 September 1859.

ULF waves are first observed on the ground and also known as geomagnetic pulsations. The solar storm of 1859 (Carrington, 1860, also known as the Carrington Event) was probably associated with a huge solar coronal mass ejection

(CME) hitting the Earth's magnetosphere and induced arguably the largest geomagnetic storm on record on September 1–2, 1859 (Stewart 1861). As we can see from Figure 2, the first geomagnetic pulsation has been recorded as quasi-sinusoidal magnetic field variations during the great magnetic storm that occurred in 1859 (Stewart 1861). Geomagnetic pulsations (Figure 2) are ultra-low frequency (ULF) plasma waves originated in the Earth's magnetosphere.

In fact, the D component of the ground magnetic field on the bottom trace mainly represents the poloidal mode ULF waves

[Wang et al, 2010, Zong et al 2017]. The magnetic field perturbation of the toroidal mode ULF waves is in the azimuth direction, and the electric field is radial perturbation usually associated with a small wave number. Whereas, the poloidal mode ULF waves are often associated with a larger wave number, and the magnetic field of the poloidal mode is radially perturbed.

Oscillations of magnetic field lines can be sustained through the collisionless plasma interaction in the magnetosphere. Whereas, the ULF waves can also be diminished when they pass through the Earth's atmosphere and ionosphere due to the ionospheric conductivity [Hughes and Southwood 1983]. In the Earth's ionosphere, due to the presence of collisional plasma and neutral atmospheric population, the oscillated magnetic field in the ULF range would be exponentially decayed via generating an additional Hall current and Pedersen current, and the direction of the magnetic field oscillation will be rotated

in 90°. Thus, the decayed ULF waves can eventually propagate to the ground in the form of electromagnetic waves.

The relationship between ULF waves in the magnetosphere and the magnetic field dissonances on the ground is as follows [Hughes and Southwood 1983]:

$$b_g/b_m \sim \Sigma_H/\Sigma_P^{(-kh)} \qquad (1)$$

where $\Sigma_P$ and $\Sigma_H$ are the height-integrated Pedersen and Hall conductivity, respectively, $b_g$ is the magnetic field on the ground and $b_m$ is the ULF field just above the ionosphere, h is the thickness of the ionosphere. As we can see from the formula, the Earth's ionosphere prefers to shield ULF waves with a large wave number k since the thickness of the ionosphere h is insensitive in time. Thus, the poloidal mode ULF waves of large wave numbers will decay significantly when they pass through the Earth's ionosphere, and it is hard to be observed on the ground. However, the toroidal mode ULF

waves usually have a small wave number, and it will be easier to pass through the ionosphere and be identified from ground magnetometer records.



ULF waves can act as important media of the magnetospheric dynamics for the mass, momentum and energy transport processes. Therefore, it is important to understand the global properties and how the energy is transported from the solar wind to the magnetosphere, ionosphere and finally the ground through the ULF wave - charged particle interactions.

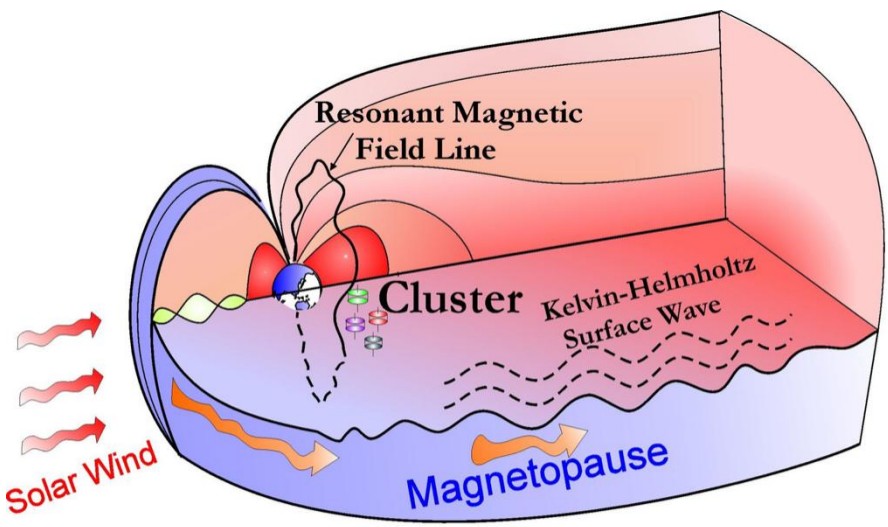


Figure 3: Illustration showing how the solar wind disturbances and Kelvin–Helmholtz surface waves excite the ULF waves and field line resonances (FLRs) in Earth's magnetosphere.

Earth's magnetospheric activities are mainly controlled by the solar wind plasma and the accompanying interplanetary
magnetic field (IMF) [e.g., Yue et al., 2009; 2010; 2011a; 2011b]. The energy coupling between the solar wind and the Earth's magnetosphere can take various forms, most often would excite different plasma waves inside magnetosphere, one of which is the ULF wave. The energy coupling between the solar wind and the Earth's magnetosphere can take various forms, most often would excite different plasma waves inside magnetosphere, one of which is the ULF wave. In 1940s, the geomagnetic signals related to the interplanetary shock impact have been identified through the ground-based magnetometer
observations and named as "Storm Sudden Commencement" (SSC) [Chapman and Bartels, 1940]. Now, it is well known as the impact of dynamic pressure impulses associated with the interplanetary shocks driven by Coronal Mass Ejections (CMEs) or Corotating Interaction Regions (CIRs).



It is now known after extensive studies that the solar wind dynamic pressure pulses (including negative and positive type) as well as the interplanetary shock can have profound effects on the magnetosphere system [Zhang et al 2010, Zong et al, 2017].  The positive or negative pressure pulses correspond to the sudden enhancements or drops of the solar wind dynamic pressures respectively and are often caused by the abrupt changes of solar wind density and/or solar wind speed. One of the typical representatives are the interplanetary shocks.

When solar wind dynamic pressure pulses impinge on the magnetosphere, the sudden raise or drop dynamic pressure will at first compress or inflate the magnetosphere. In the meantime, the fast magnetosonic waves will be launched inside the magnetosphere, and then standing ULF waves usually will be formed subsequently in the magnetosphere, occasionally even inside the plasmasphere [e.g. Zong et 2009, Zhang et al 2010, Liu et al 2010], thus transporting the energy of solar wind into the magnetosphere. The generation mechanisms of different dayside ULF waves can be distinguished by their preferable occurring region. The K-H instability mechanism needs a shear flow to meet the instability threshold condition, therefore the main occurring region are the dawn and dusk flank side of the magnetopause. The dynamic pressure pulses on the contrary are responsible for the dayside local noon region (Figure 3).

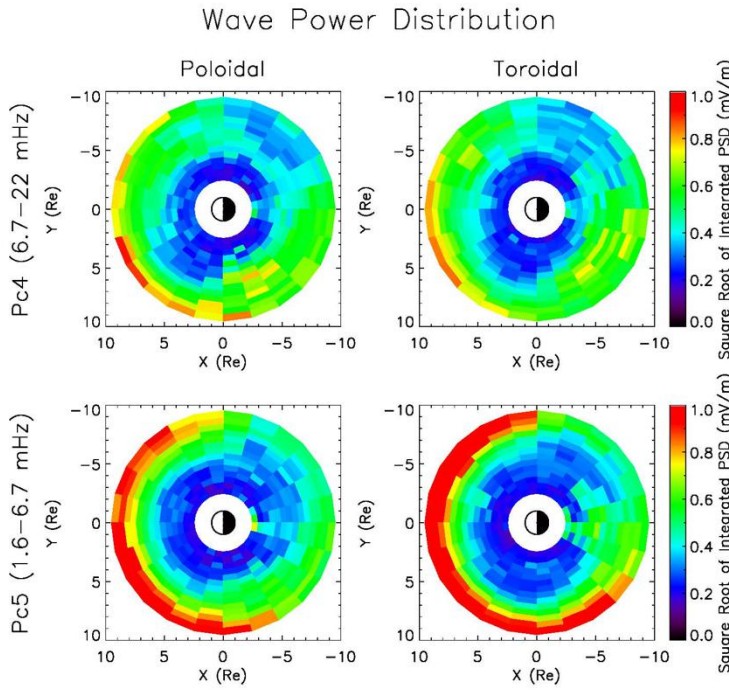

Figure 4: The equatorial distribution of ULF wave power investigated based on electric field measurements of 12 years (2008.1.1-2019.12.31) from THEMIS (Liu et al., 2009). The square root of integrated power spectral density of azimuthal

electric field (Ea, poloidal mode) and radial electric field (Er, toroidal mode) in Pc4 and Pc5 frequency ranges are averaged in each bin.

Figure 4 shows the responses of both poloidal mode and toroidal model ULF waves to the solar wind forcing at different magnetic local time sectors. It is suggested that Pc4 and Pc5 ULF wave power is mainly supplied from external solar wind

sources, i.e. solar wind forcing (Liu et al. 2009, 2010). As is shown in Figure 4, the distributions of the wave power (square root of integrated power spectral density) of the azimuthal electric field (Ea, poloidal mode) component are averaged from 12 years THEMIS data sets. The wave power is found to be stronger in the dayside magnetosphere compared to that in the nightside. Also, the wave power in the pre-midnight is larger than that in the post-midnight region. The wave power is observed dominantly at higher L shells, which show the consistency with the scenario that the poloidal mode Pc4 and Pc5

ULF wave generally have external sources - solar wind forcing including interplanetary shocks, solar wind positive and negative dynamic pressure pulses (e.g., Zong et al. 2009; Zhang et al. 2010; Liu et al. 2010).

The study of magnetospheric response to solar wind forcing related to a sudden change in solar wind dynamic pressure has at least two obvious advantage points: the magnetospheric response to sudden change of the solar wind dynamic pressure will generate significant and easily identified electromagnetic signals; and the energy source for excited ULF waves is rather

clear without temporal ambiguity. Thus, in the present paper, based on the ULF wave – charged particle interactions, I will focus on how the magnetosphere response to solar wind forcing -- the solar wind dynamic pressure pulses including positive and negative ones.

## 1.3 Charged particles in the inner magnetosphere





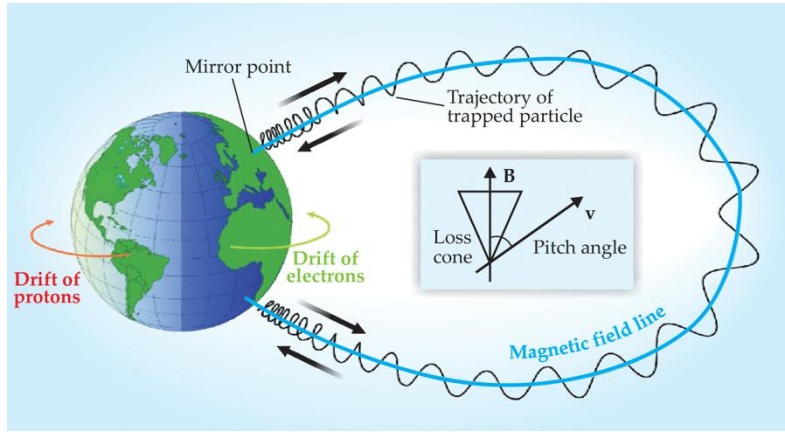

Figure 5: A schematic view of the charged particle motion in the Earth's inner magnetosphere. Particle's gyro motion

around field lines, "bounce" back-and-forth motion and drift motion due to the gradient and curvature of the magnetic field.

Ions drift towards west and electrons drift east, and thus generate the ring current, an electric current circulating around the

Earth.

The inner magnetosphere includes radiation belt, ring current and plasmasphere,  which are three overlapping regions

with energy of their particle population quite different.[Yue et al., 2017a, 2017b]. Van Allan radiation belt is composed of

energetic particles with energy greater than 100keV, whereas the ring current contains mainly energetic ion species,

(hydrogen, helium, and oxygen) of tens keV to about 400keV. The ring current and Van Allan radiation belt are overlapped

in space with cold plasmaspheric population (typically a few eV).

The plasma density in the magnetosphere controls the time scale in response to the solar wind forcing. The mass density

is one of the key parameters for the Alfvén speed which determines the magnetospheric response to the low-frequency ions

variation in the ULF wave range, whereas the background electron density dominates the electrons oscillating in VLF and

radio wave range. Thus, the mass density is one of the controlling factors for the radiation belt and  ring current dynamic

process.




Charged particles in the Earth's magnetosphere will experience three kinds of periodical motions corresponding to three different invariants: gyrating around magnetic field lines; bouncing back and forth along the field line between "mirror points" located at lower altitude, and drifting across field lines due to the electric field as well as gradient and curvature of the magnetic field lines, see Figure 5. When the charged particles moving in the inner magnetosphere, the time scales for the three kinds of motion can be estimated with the dipole magnetic field:

$$
\begin{aligned}
T_c &= (0.66\,\mathrm{s})\frac{100\mathbf{nT}}{B}A \\
T_b &= (5\,\mathbf{min})\left(\frac{l_0}{10R_E}\right)\left(\frac{\mathbf{keV}}{W_\parallel}\right)A^{\frac{1}{2}} \\
T_D &= (56\,\mathbf{Hour})\left(\frac{r}{5R_E}\right)^2\left(\frac{B}{100\mathbf{nT}}\right)\left(\frac{\mathbf{keV}}{W}\right)
\end{aligned}
\tag{2}
$$

where A is the mass ratio of the particle to the proton, W is the particle's kinetic energy, $l_0$ is the length along the magnetic field line between two mirror points at the northern and southern hemisphere, r is the distance to the Earth's centre from the equator and B is the magnitude of the magnetic field. The representative values are given in Table 1.

The dynamics of radiation belt and ring current are strongly governed by the interactions between different charged particle populations that are coupled through the variation of all kinds of electromagnetic waves and wave - particle interactions. The mentioned three invariants do not always keep constant. The violations of these invariants may result from interactions with variations of the magnetic field and electric field when the time scale of the electromagnetic field disturbances is comparable to the three kinds of periodical motions - drift, bounce or gyration frequency of the particles.

**Motion Frequencies of Charged Particles in Magnetosphere (L=4.5, $\alpha_{eq}$=60°, B=350 nT)**.





| Regions | Particles | Gyration | Bounce | Drift |
|---|---|---|---|---|
| **Plasma-sphere** | e⁻ (10 eV) | 61.5 kHz | 20.0 mHz | 1.6E-5 mHz |
| **Ring Current** | O⁺ (100 keV) | 2.1 Hz | 38.0 mHz | 0.16 mHz |
| | H⁺ (100 keV) | 33.5 Hz | 151 mHz | 0.16 mHz |
| **Radiation Belt** | e⁻ (1 MeV) | 61.5 kHz | 3.0 Hz | 1.6 mHz |

Besides the magnetic reconnection, the solar wind energy can be also transported into the magnetosphere, the ionosphere
and finally the ground through ULF waves. The mechanisms concerning how ULF waves interact with charged particles in
the magnetosphere and their involvement in energy transporting process would be addressed by  spacecraft constellation
observations and particle simulations. It will help us understand the complex but fundamental problems of mass, energy and
momentum transport processes in the magnetosphere,  and have  a wide  range of applications in space weather.

The paper is based on my Hannes Alfven Medal lecture at the EGU General Assembly 2020, and organized as follows with
emphasis on Sects. 2, 3, 4 and 5.





# 2.0 Magnetospheric response to solar wind forcing

## 2.1 "One kick" scenario

Charged particles in the Earth's magnetosphere can be significantly affected by the impact of the passage of an interplanetary shock [Matsushita et al, 1961, Brown et al, 1961, Ullaland et al, 1970]. Enhanced precipitation of ~10s keV electrons into the Earth's atmosphere has been observed immediately which last up to ~10 minutes when an interplanetary shock impacts on the geospace system [Su et al., 2011; Yue et al., 2013].

The sudden changes of charged particle fluxes in the inner magnetosphere including both relativistic electrons in the radiation belt [Arnoldy 1982, Blake et al, 1992, Li et al, 1993, Hudson et al, 1994, Tan et al, 2004, Zong et al., 2009, Hao et al., 2019] and energetic ions [Zong et al, 2012,l 2017; Ren et al. 2016, 2017a] in the ring current region are noted to be closely related to SSCs (sudden storm commencement) caused by the interplanetary shock impacting on the Earth's magnetosphere. These results suggest that a significant portion of energetic charged particles in the ring current and radiation belt and region could be produced even before the build-up of the enhanced ring current which produces the magnetic storm.

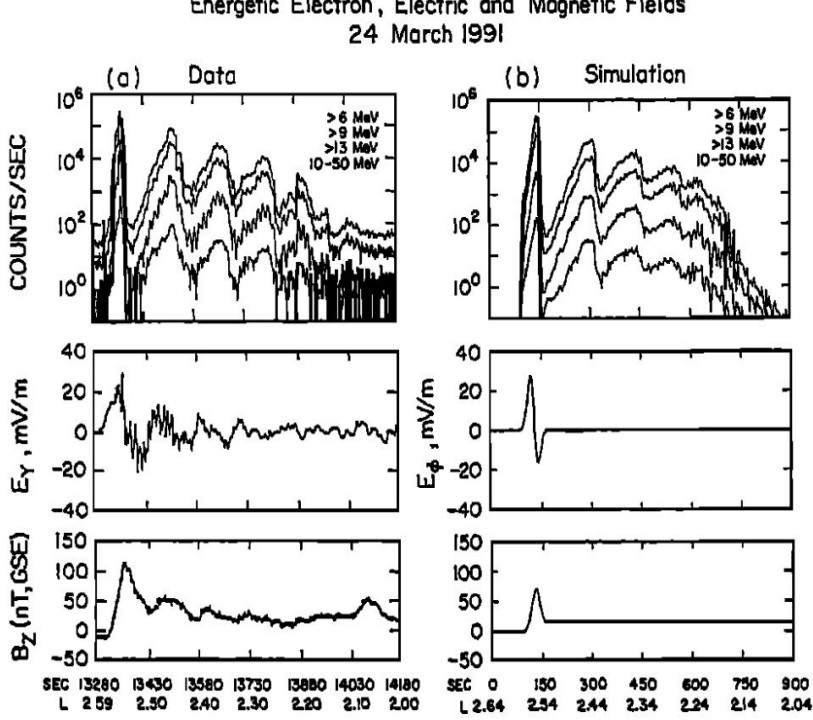

Figure 6: Satellite observation and Simulation comparisons *at the time of the March 24, 1991 SSC. Panels show four energetic electron channels measurements over 10-50 MeV, the measured electric field and the Bz magnetic field component, and the simulated results in the same format at a spatial location corresponding to the trajectory of the CRRES satellite [Li et al, 1993].*

Energetic particles of both electrons and ions up to 15 MeV have been observed in the radiation belt due to the impact of a strong interplanetary shock on the Earth's magnetosphere on 24 March, 1991 [Blake et al, 1992]. It is believed that both relativistic ions and electrons are accelerated quickly by an induced electric field pulse generated by the passage of the interplanetary shock [Li et al, 1993, Hudson et al, 1994]. A rapid (a few mins) formation of a new electron radiation belt at $L \simeq 2.5$ were observed in the slot region besides the inner and outer radiation belts, which lasted for a few years [Blake et al, 1994].

Assumed that a running pulse with a bipolar electric field has been generated inside the magnetosphere by the compression and relaxation of the Earth's magnetosphere caused by the interplanetary shock impinging on the Earth's magnetosphere. As





shown in Figure 6, test particles interacting with this assumed asymmetric bipolar electric field pulse ("One Kick") caused by the passage of the interplanetary shock has been proposed to explain the newly formed electron radiation belt at L ≃ 2.5 [Li et al, 1993, Hudson et al, 1994]. This simulation has shown that a few MeV energetic electrons at L > 6 could be energized up to 40 MeV and be radially transported to L ≃ 2.5 during a fraction of their azimuthal drift period. The

simulation results can reproduce the observed very energetic electron injection and their drift echoes. The acceleration process can be understood as that the first adiabatic invariant is conserved (adiabatic acceleration) and the electrons are accelerated by the assumed single bipolar electric field pulse. The time scale of acceleration processes is about1 min since the electromagnetic pulse would be running away in that time period. This is so called "One Kick" scenario of an interplanetary shock interacting with the Earth's magnetosphere.

Since then, extensive test particle and MHD simulations have been carried on to study the particle acceleration related to the interplanetary shock impact [e.g. Hudson et al, 1995, Kress et al, 2007]. It has been pointed out [Friedel et al., 2005], the "One Kick" model was capable to reproduce some observational features for the event on March 24, 1991. However, it seems that the model can explain only this sole shock event and is not applicable for other interplanetary shock events in the magnetosphere. Thus, it remains unsettled how shock-related energetic particles are created and accelerated in the

magnetosphere [Friedel et al. 2005, Baker et al, 2004].

## 2.2    Poloidal ULF wave - charged particle interaction scenario

In the magnetosphere, the energetic charged particles are mainly drifting in the azimuthal direction, with electron drifting eastward and ion drifting toward west. The electric field of poloidal mode ULF waves is also lying in the azimuthal direction. When both the drift direction of charged particles and the propagating direction of ULF waves are the same, the electric field

carried by poloidal ULF waves would accelerate/decelerate the drifting charged particles. However, it should be noted that only those resonant electrons with a drift speed  of approximately the wave propagation speed of the poloidal mode ULF wave, could gain energy constantly. Charged particles, bearing both the acceleration and the deceleration processes would cancel out with a relatively small energy gain during one wave period.

As we have already shown in the introduction, ULF waves in the Earth's magnetosphere could be excited by the impinge

of positive or negative solar wind dynamic pressure pulses. Energetic charged particle fluxes modulated by ULF waves in the Pc 5 band were found by Brown et al. [1961] at first. A close correlation between the charged particle flux variations and the intensity of ULF waves has been found for both case studies (e.g., Tan et al, 2004, Zong et al 2007) and statistic surveys (e.g., Rostoker et al, 1998, Mathie and Mann, 2001, O'Brion et al, 2003).

Due to the comparable periods between the drift and bounce motions of the charged particles and the ULF waves in inner

magnetosphere, drift resonance or drift-bounce resonance may be satisfied. One to one correlation between ULF wave cycles



and fluctuations of charged particle fluxes have been found which indicates ongoing wave-particle interactions, and the interactions would accelerate the magnetospheric particle significantly ([Zong et al 2007, 2009, 2017].

Now, tremendous efforts have been made to understand how the interplanetary shocks affect the charged particles in radiation belt and ring current region. By using observations from the Cluster and Double Star constellation, it has been found that after solar wind dynamic pressure pulses impinging upon the magnetosphere, the acceleration of radiation belt energetic electrons could start immediately and can last for up to a few hours [Zong et al, 2009, Zong et al, 2012, Zong et al, 2017]. The prime acceleration mechanisms are drift-resonance or drift-bounce resonance with ULF waves excited by the interplanetary shock impacting on the magnetosphere.

A direct observation of such a ULF wave – charged particle interaction scenario is shown in Figure 7. The onset of strong ULF waves is associated closely with a strong interplanetary shock impact on the magnetosphere on 7 November, 2004. At the same time, the energetic electrons are accelerated quickly and directly one to one correlated with the shock-induced ULF waves. As shown in Figure 7, at 18:27 UT on November 7, 2004, an interplanetary shock with a maximum dynamic pressure of ~ 70 nPa has hit the magnetosphere. At the same time, large amplitude ULF waves with electric field of ~40mV/m have been observed when Cluster spacecraft fleet moving in the morning side of the plasmasphere. The ULF waves are excited by the IP shock impinging on the magnetosphere. The one to one correlations between the flux variations of energetic electrons and the ULF wave oscillations suggest that the shock-induced ULF waves causes the observed charged particle acceleration.

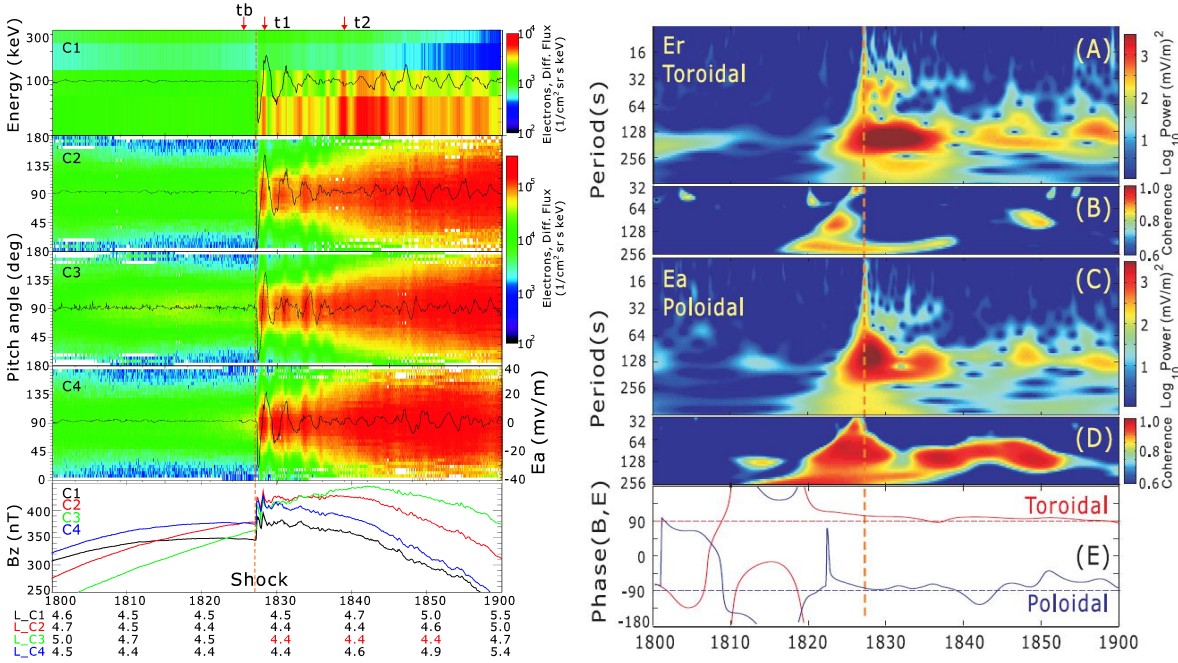



Figure 7 Left: from top to bottom, (a) the energetic electron spectrum; the pitch angle distributions overplotted with azimuthal electric field Ea (black line) in the mean-field-aligned (MFA) coordinate system, and (e) the magnetic field

Bz components. The dashed vertical line shows the interplanetary shock arrival time. Right: Measurements from Cluster C3. (a) continuous wavelet power spectrum of the radial electric field, (b) the squared wavelet coherence between the radial electric field and the integrated energetic electron flux, (c-d) the same for the azimuthal electric field. (e) Phase difference between electric fields and magnetic fields for the toroidal mode (red) and poloidal mode (blue), indicating both poloidal and toroidal modes are standing waves [Zong et al,2009].


    With an amplitude as high as 40 mV/m, the electric field of poloidal ULF waves on the charged particle drift path can double the energy of electrons with a few hundred keV in only several wave periods. This is much faster than other acceleration processes e.g. gyro-resonances via VLF waves, suggesting that the observed ULF waves are sufficient to explain the observed electron acceleration through drift resonance.

The toroidal and poloidal modes have similar wave powers, however, coherences between the electric fields for both poloidal and toroidal mode ULF waves and the integrated energetic electron flux are rather different based on the wavelet technique [Grinsted et al., 2004]. The high coherences of above 0.9 appears continuously and only with the poloidal mode ULF waves. This confirms the energetic electrons are accelerated predominately by poloidal wave-carried electric field.

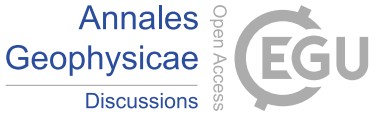

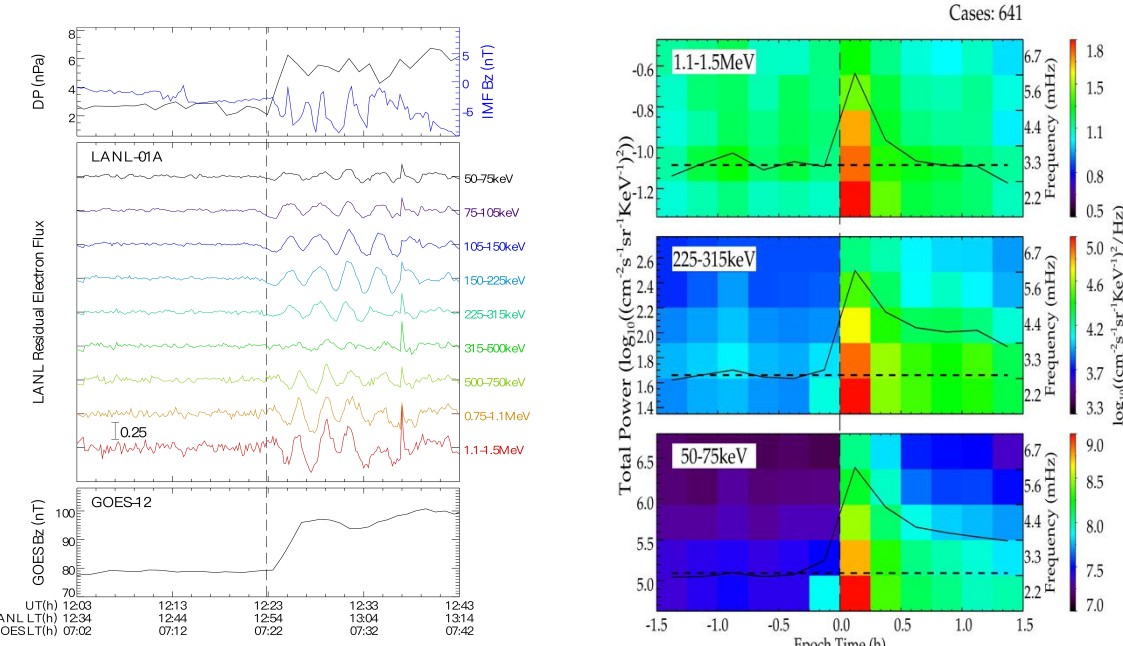

Figure 8. Left: The response of the magnetic field and electron fluxes at geosynchronous orbit to an IP shock on 29 May 2003. (a) Shifted interplanetary magnetic field (IMF) and solar wind dynamic pressure observed by ACE; The vertical dashed line indicates the shock arrival time at geosynchronous orbit. Energetic electron fluxes measured by LANL satellite (LANL-01A); the response of the geosynchronous magnetic field observed by GOES 12. Right: Superposed epoch analysis of 641 IP shock related dynamic power spectra of the electron fluxes at geosynchronous orbit. (bottom to top) Median value of dynamic power spectrum of the electron fluxes for nine channels (50–75 keV, 225–315 keV, and 1.1–1.5MeV). Epoch time zero is denoted by the black vertical dashed line. [After Liu and Zong, 2015].

This scenario has been further examined by systematically statistics study based on geosynchronous energetic particle observations for 215 interplanetary shock events during 1998–2007 [Liu and Zong, 2015]. It is shown that electron fluxes with an energy less than ~300 keV increase after the shock impact whereas electron fluxes with an energy higher than ~300 keV show smaller increases, become unchanged or even decrease eventually at geosynchronous orbit (Figure 8). The electron flux oscillations following the shock arrival have also been investigated. Statistical analyses revealed a frequency preference for energetic electron flux oscillations of 2.2 mHz and 3.3 mHz [Liu and Zong, 2015]. The compressional effect of IP shocks can cause acceleration due to both magnetic field magnitude enhancement and related azimuthal electric field. The electron fluxes increasing or decreasing are depended on the pre-conditional phase space density profile. The energy



change of electrons is attributed to the compressional effect of interplanetary shocks and the interaction with shock-induced ULF waves.

It is also indicated that energetic electrons with low-energy (high-energy) will resonate with high-m (low-m) ULF waves and have different modulation features. The results show the magnetospheric response to ULF waves excited by the interplanetary shock impact from the energetic particle point of view.

In brief, the interplanetary shock related to energetic electron acceleration in radiation belt starts almost immediately following the shock arrival. The acceleration process includes two contributing steps, the first acceleration is related to the initial magnetospheric compression by the interplanetary shock impact and then immediately followed by drift-resonant or drift-bounce-resonant acceleration by poloidal ULF waves excited by the passage of the interplanetary shock [Zong et al, 2009, Zong et al, 2012, Zong et al, 2017]. This is the shock induced ULF waves – particle interaction scenario. Such a

scenario on shock induced ULF waves' interaction with charged particles has been further confirmed by many other satellite observations [Clauderpierre et al, 2013, Foster et al, 2015, Korotova et al, 2018].

## 3. Generalized theory on the drift resonance

In this section, the traditional drift resonance theory will be introduced first, then the generalized drift resonance theory on charged particles resonating with growth and damping ULF waves and charged particles resonating with azimuthal localized

ULF waves will be described. Finally, I will show how radiation belt relativistic electrons resonating with localized growth and damping ULF waves in detail.

In the magnetosphere, the frequencies of ULF waves are comparable to the frequency of charged particle's drift or bounce motion. Analogous to gyro-resonance, it is suggested in the 1960s that charged particles trapped by the Earth's magnetic

field could resonantly interact with ULF waves standing on a field line through the particles' bounce and drift motions (Dungey 1964, Southwood 1969).

The drift-bounce resonance condition is written as:

$$\omega - m \cdot \omega_d = N \cdot \omega_b \qquad (3)$$



where N is an integer (normally 0, ±1, ±2), m represents the azimuthal wave number, and ⍵, ⍵$_d$ and ⍵$_b$ are wave frequency,

particle's drift and bounce frequencies, respectively. Since ⍵d and ⍵b depend on the charged particle's energy, for a given location, the resonance energy can be determined in theory if the ULF wave's frequency is known.

The charged particles are moving in the electric field carried by the ULF waves during their drift-bounce motions, thus, their energy can be accordingly changed. The energy change rate of a charged particle interacting with poloidal mode ULF waves can be written as [Southwood and Kivelson 1982,1984]:

$$\frac{dW}{dt} = \mu \frac{\partial B_P}{\partial t} + qE \cdot V_d \qquad (4)$$


where $\frac{dW}{dt}$ , E, V$_d$, and $\mu$ are the change rate of the particle energy, the wave electric field, the particle drift velocity and the particle magnetic moment, respectively. The subscript p denotes the component parallel to the background magnetic field.

For energetic electrons resonant with ULF waves, the bounce frequency is usually much higher than the wave frequency and the particle's drift frequency [Zong et al 2009]. Therefore, charged particles' interaction with ULF waves via the drift-

bounce resonance can only be excited at N = 0 (Fundamental mode), as shown in Figure 9. In this way, the drift-bounce resonance degenerates to the drift resonance, the bounce motion has no relationship with the ULF - particle interaction.

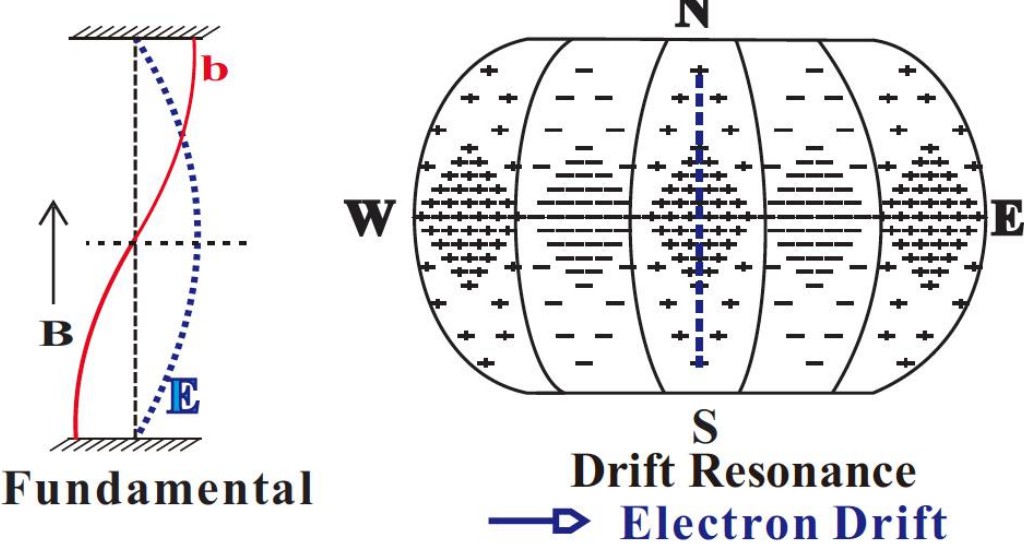



*Figure 9. Fast Acceleration of Electron by resonating with Poloidal ULF Waves. A schematic of N=0 drift resonance in a fundamental mode standing wave. The eastward electric fields are indicated by plus and minus, and their magnitudes*
*correspond to the density of the symbols.*

$$\omega = m \cdot \omega_d \qquad (5)$$

Once the drift resonance is satisfied, the resonant electrons with fundamental mode ULF waves seem to be stagnant azimuthally in the ULF wave moving frame. Thus, the resonant electrons can be accelerated very quickly since only uniform electric field can be experienced by the resonant electrons. Resonating with the fundamental poloidal ULF waves is a very
efficient way to accelerate electron in the radiation belt region since the electric field of poloidal ULF waves is the same as the charged particle drift direction [Zong et at 2009, 2017, Hao et al, 2019].

### 3.1 Generalized drift resonance with growth & damping ULF waves

In the traditional drift resonance theory, the ULF wave growth rate is assumed as time independent, positive, and the
amplitude of the ULF wave is extremely small. This is not agreed with satellite observations in the magnetosphere, and the interplanetary shock induced ULF waves usually have huge amplitudes and experience growth (a positive growth rate) and damping (a negative growth rate) stages [Zong et al, 2009, Zhang et al 2010, Liu et al 2010]. Thus, a more generalized theory dealing with the interaction between ULF waves' and charged particles in the magnetosphere for a time dependent ULF wave evolution is required.



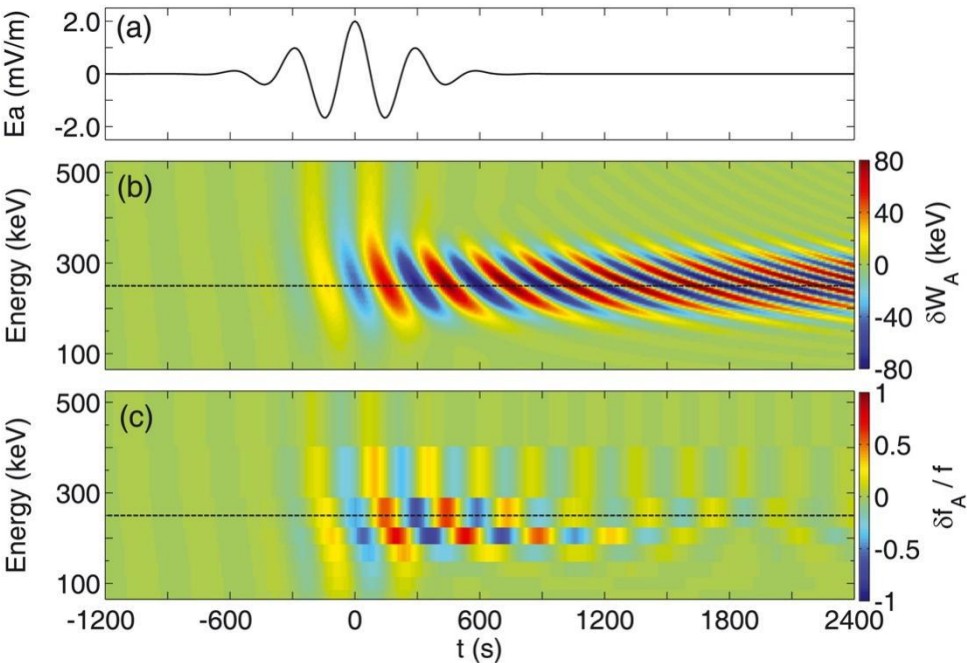

Figure 10: Energetic Electrons' interaction with a ULF wave during its growth and damping stages. (a) ULF wave-associated electric field in the azimuthal direction, (b) electron energy gain from the ULF wave as a function of time and energy, with the resonant energy of 250 keV represented by the dashed line, and (c) predicted spectrum of electron residual phase space densities observed by a Magnetic Electron Ion Spectrometer (MagEIS)-like particle detector with finite time and energy resolution [Zhou et al, 2016].

A drift resonance theory with growth and damping ULF waves has been developed [Zhou et al, 2016, Zong et al, 2017]. In there, a time dependent imaginary wave frequency has been adopted to describe the growth and damping of the waves in the generalized drift resonance theory, therefore, the interactions between charged particles and growth and damping ULF waves can be studied [Zhou et al, 2016, Zong et al, 2017].

The generalized drift resonance theory with growth and damping ULF waves, it allows a time - dependent ULF wave growth rate, which is large and positive in the wave leading growth phase and decreases to negative values gradually in the damping phase. This assumption is based on ULF waves excited by the interplanetary shock impact on the  magnetosphere



[Tan et al, 2004, Zong et al, 2009, Zhang et al, 2010, Clauderpierre et al, 2013, Foster et al, 2015, Korotova et al, 2018, Hao et al, 2019].

The wave-associated electric field can be given by:

$$\boldsymbol{E} = E_\phi e^{-\frac{t^2}{\tau^2}} e^{i(m\phi - \omega_r t)} \boldsymbol{e}_\phi, \qquad (6)$$

where $\phi$ is the magnetic longitude (increasing eastward), $\omega_r$ is the real part of the wave angular frequency, and $m$ is the
wave azimuthal wavenumber. Equation (6) describes a Gaussian amplitude envelope of the electric field oscillation.

The change rate of particle's kinetic energy within the waves is given by:

$$\frac{dW}{dt} = q\boldsymbol{E} \cdot \boldsymbol{v_d}, \qquad (7)$$

where $W$ is particle's kinetic energy, $q$ is particle's charge, and $\boldsymbol{v_d}$ is the magnetic gradient and curvature drift velocity. In
the terrestrial dipole field, it is approximated by

$$\boldsymbol{v_d} = -\frac{\gamma + 1}{2\gamma} \frac{6L^2 W}{q B_E R_E} \left(0.35 + 0.15 sin\alpha_{eq}\right) \boldsymbol{e}_\phi. \qquad (8)$$

where $R_E$ is Earth's radius, $B_E$ is the equatorial magnetic field on Earth's surface, L is the L shell parameter, and $\gamma$ is the
relativistic Lorentz factor. For a nonrelativistic, equatorially mirroring particle, $\boldsymbol{v_d}$ can be rewritten as

$$\boldsymbol{v_d} = -\frac{3L^2 W}{q B_E R_E} \boldsymbol{e}_\phi. \qquad (9)$$


Therefore, the change rate of kinetic energy can be rewritten as

$$\frac{dW}{dt} = -\frac{3L^2 W}{B_E R_E} E_\phi e^{-\frac{t^2}{\tau^2}} e^{i(m\phi - \omega_r t)} \qquad (10)$$





$$\delta W = -\frac{\sqrt{\pi}}{2}\frac{3L^2 W}{B_E R_E}E_\phi \times k(\tau) \times 2 \times e^{im\phi} \times e^{-im\omega_d t}$$

(11)

which indicates the frequency of $\delta W$ is $m\omega_d$ rather than $\omega_r$.


As we can see from Figure 10, with the wave amplitude increasing, the electron flux PSD is oscillating with a gradual

enhancement, and the phase difference between difference electrons with a lower and a higher energy is changing from a

small value to ~ 180° when the amplitude of the ULF wave stops growing. Whereas, In the ULF wave damping stage, both

the variations of energetic electron PSD and the phase shift between electron fluxes with different energies continue to

increase till the phase mixing effect attenuates the particle PSD oscillations. A distorted energy spectrum can be expected as

the results of energetic electrons resonating with a growth and damping ULF wave.

Resonant charged particles signatures can be explained by the generalized theory whereas equations in the traditional drift

resonance theory are invalid. It is found that the distorted energy spectrum predicted from the generalized theory for the

interactions between charged particles and growth and damping ULF waves are very well in agreement with observations

from Van Allen Probes. Thus, the generalized theory for drift resonance with growth and damping ULF waves can provide

new insights into the interactions between ULF waves and charged particles in the magnetosphere.

### 3.2  Generalized drift resonance with localized ULF waves: "Boomerang" pitch angle distribution

ULF waves in the traditional drift resonance theory are considered to be global distributed, and the amplitude of ULF

waves is azimuthally uniform, i.e. independent of magnetic longitude (magnetic local time). However, the observations have

suggested there may be localized ULF waves. Pitch angle distribution of boomerang stripes is found to be the result of drift

dispersion among resonant charged particles interacting with localized ULF waves in an azimuthally distant region of the



magnetosphere [Hao et al, 2017, Zong et al, 2017]. Therefore, we have introduced a magnetic longitude dependence of the
ULF wave amplitude into generalized drift resonance with localized ULF waves.

As shown in Li et al [2017], the von Mises function is adopted to study the effect of a localized ULF wave (magnetic longitude dependence) in the ULF wave – charged particle interaction. The spatial localized ULF waves described here are transverse, poloidal ULF waves with azimuthal electric field oscillations. Whereas, the temporal evolution of these ULF
waves are the same as the traditional one [e.g. Southwood and Kivelson, 1981], i.e., the wave is time independent and very small in terms of the wave magnitude.

The electric field of the localized ULF waves is given by

$$\boldsymbol{E}(t, \phi) = \frac{E_{\phi 0}}{2\pi \, I_0(\xi)} \exp\left[\xi \cos(\phi - \phi_0)\right] \cdot \exp i\left(m\phi - \omega t\right) \hat{\boldsymbol{e}}_{\phi}, \tag{12}$$

where $m$ is the ULF wave number, $\phi$ is the magnetic longitude (increasing eastward), and $\omega$ is the ULF wave angular
frequency.

Thus, the average rate of the particle energy gain from the transverse ULF waves is  [Northrop, 1963]

$$\frac{\mathrm{d}W_A}{\mathrm{d}t} = \frac{q\, v_{d\phi} E_{\phi 0}}{2\pi \, I_0(\xi)} \exp\left[\xi \cos(\phi - \phi_0)\right] \cdot \exp i\left(m\phi - \omega t\right), \tag{13}$$

where the subscript A denotes the average over many gyration periods, q is the particle's charge, and $V_{d\phi}$ is the azimuthal component of the particle's drift speed.


The azimuthal drift speed $V_{d\phi}$ of the particle can be obtained from following equation if an equatorially mirroring particle is considered in the dipole field

$$V_{d\phi} = -3L^2 \, W / qB_E \, R_E \tag{14}$$

where $L$ is the $L$ shell number, $R_E$ is Earth's radius, and $B_E$ is the magnitude of the equatorial magnetic field.

Then, the energy gain $\delta W_A$ from the ULF waves can be obtained if we integrate $dW_A/dt$ along the particle's unperturbed drift orbit.



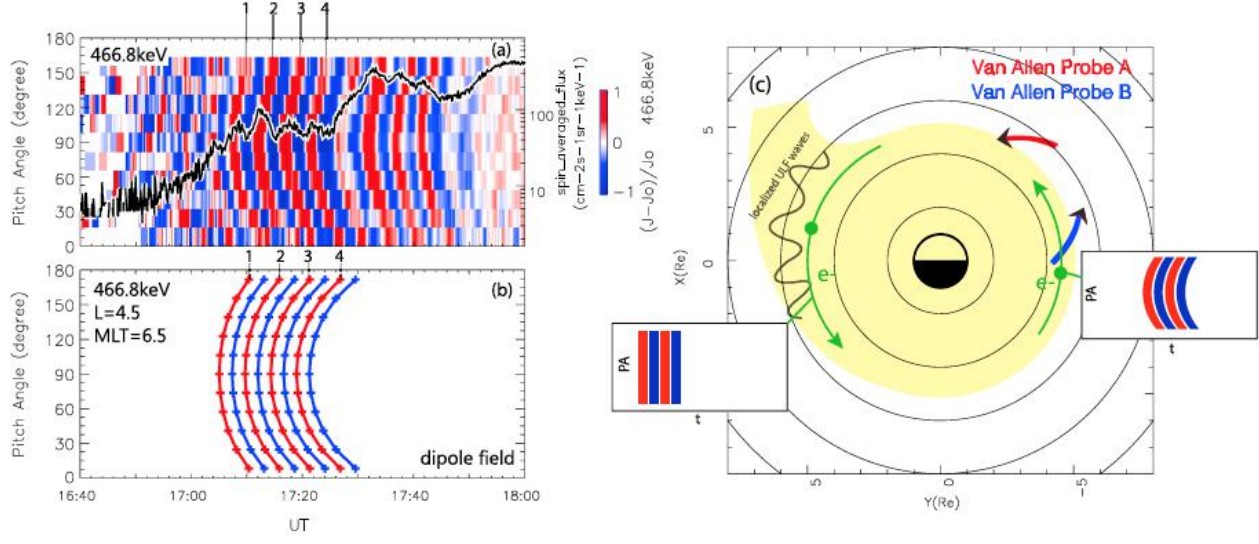

Figure 11: *Comparison between observed boomerang stripes and pitch angle dispersion by time-of-flight backward tracing*
*method. (a) Pitch angle evolution of 466.8 keV electrons. (b) Arrival time estimation for peaks and valleys of 466.8 keV*
*electrons with pitch angle varying from 5° to 175°. Electron drift velocity is calculated in terrestrial dipole field with*
*relativistic effect included. (c) A diagram for the pitch angle evolution of electrons interacting with localized ULF waves.*
*The plasmasphere is indicated in yellowish.*

Keeping these in mind, the generalized drift resonance theory of particles' interaction with localized ULF waves are
applicable for the event mentioned in Li et al. [2017] and for the pitch angle evolution of "Boomerang-shaped" by Hao et al.
[2017] which used Chinese navigation satellite and Van Allan probes data. The "Boomerang-shaped" pitch angle evolutions
of relativistic electrons appear immediately after an interplanetary shock impinges on the magnetosphere on 7 June, 2014 as
shown in Figure 11. The observed electron flux at different pitch angles is strongly modulated by ULF waves excited by the
interplanetary shock impact.



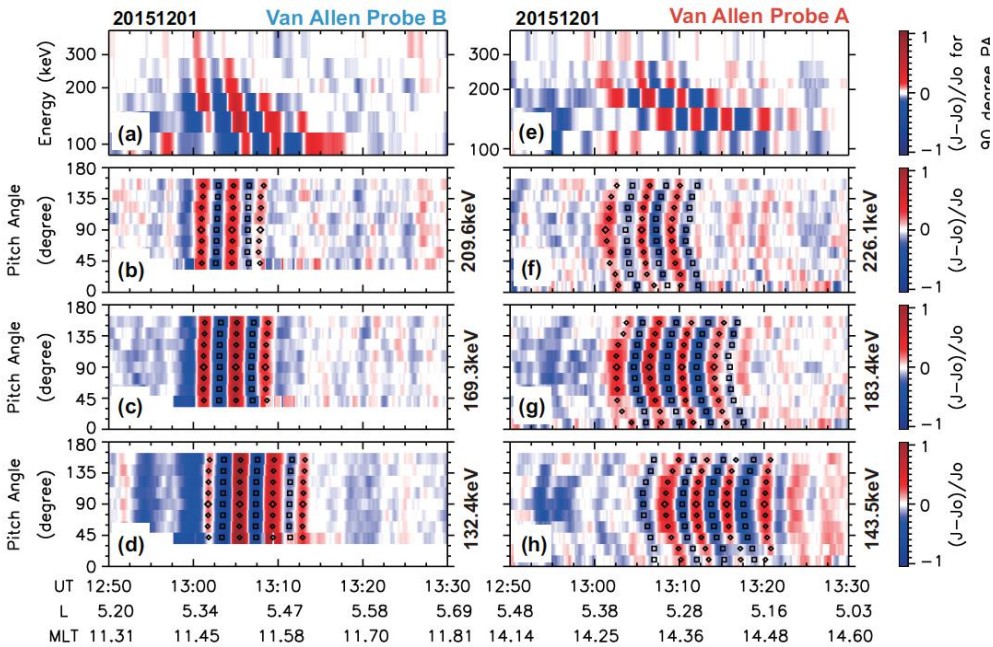

Figure 12: Observed data of electrons from MagEIS-A and MagEIS-B on 1st December 2015. (a): Residual flux profile in
energy-versus-time plot from MagEIS-B.(b-d): Residual flux profile in pitch angle-versus-time plot of energy 209.6keV,
169.3keV

and 132.4keV. Rhombuses and squares are maximum and minimum points on residual flux of stripes. (e): The same as (a)
but from MagEIS-A in the same time interval. (f-h): The same as (b) but for 226.1keV, 183.4keV and 143.5keV measured from
MagEIS-A *[Zhao et al, 2020]*.

As demonstrated in Figure 11, shock-induced ULF waves are suggested to be confined in a limited azimuthal region
(possibly the plasmaspheric plume), which is westward of the Van Allen Probe spacecraft. Then, ULF wave-modulated
energetic electrons drift out of the ULF wave – charged particle interaction region before they are observed by the distant
spacecraft. The drift speed of the modulated energetic particle is depending on its energy and pitch angle. The difference in
energy and pitch angle of the energetic electrons would produce a drift dispersion, i.e. equatorially mirroring 90° pitch
angle electrons would drift faster and be observed first. This effect will lead to distorted particle pitch angle stripes to form
"Boomerang-shaped" evolutions in pitch angle spectra for each electron energy band. The observed boomerang stripes as
well asmodulations in the electron energy spectrogram can be reproduced by using by time-of-flight backward tracing
method [Hao et al, 2017, Zong et al, 2017].

Furthermore, ULF wave - radiation belt electron drift resonance can be depicted by quasi-periodic stripes, either straight or "Boomerang-shaped", in the pitch angle spectrum of electron fluxes as shown in Figure 12 and Figure 13. Boomerang-shaped stripes on pitch angle distribution are evolved from straight ones after resonant electrons drift away from the azimuthally localized ULF wave - particle interaction region. Also, it provides a new method based on the time-of-flight

tracing technique to identify the region of ULF waves interacting with particles. Thus, , it is crucial to take both the spatial distribution and temporal evolution of ULF waves into consideration for both drift resonance and drift-bounce resonance [Zhao et al, 2020].

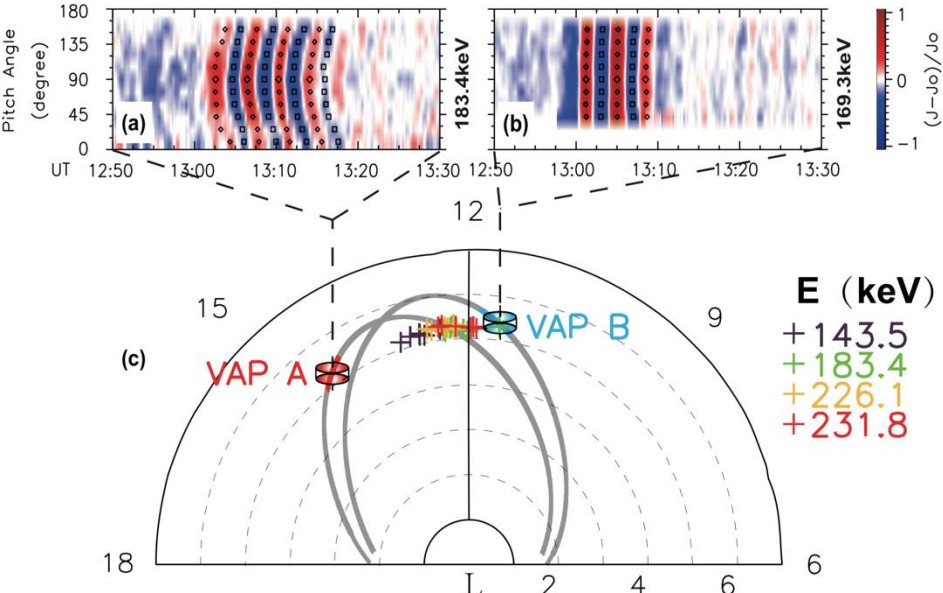

Figure 13: Pitch angle distribution observed by VAP-A and VAP-B. (c): The source places and Van Allen Probes orbits in the

equatorial plane. These cross-shaped symbols represent the results by the time-of-flight backward tracing method from VAP-A. The local time labelled is magnetic local time, while numbers on x-axis are L shell [Zhao et al, 2020].

The study of "Boomerang-shaped" evolutions in pitch angle spectra would tell us not only that where the drift resonance is taking place, but also the possible scale size of the ULF wave - particle interaction at a location distant away from the

spacecraft. These results add new understanding to the radiation belt dynamics.



# 3.3 Radiation belt "relativistic electrons" acceleration by drift resonance

What will happen if charged particles are in drift resonance with both growth and damping ULF waves and localized ULF

waves? An excellent example is given in the Figure 14, relativistic energetic electrons resonating with localized growth and

damping ULF waves can lead to very rapid ultra-relativistic electron acceleration in the radiation belt region.

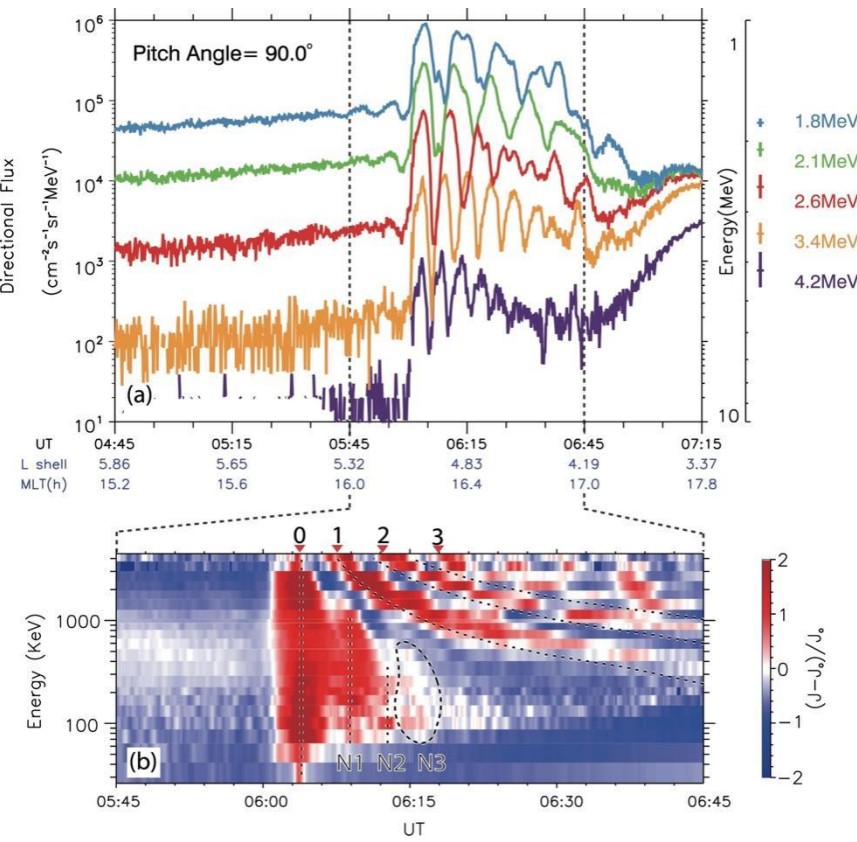

Figure 14: The response of electrons to the storm sudden commencement. (a) 90◦ relativistic and ultra-relativistic electron flux

measured. The width of each energy channel is plotted at the right of the panel. (b) Residual flux (J–J0/J0) of 90◦ electrons in

energy-versus-time plot. Dashed lines with numbers 1, 2, and 3 give the predicted energy-dependent arrival time of the first,

second, and third drift echoes of the initial flux enhancement (as marked with the vertical dashed line 0). N1, N2, and N3



indicate series of ultralow frequency modulation to the population not in resonance with the m = 1 mode ultralow frequency
wave, of which the arrival time does not match the prediction of drift echoes. MLT = magnetic local time [Hao et al, 2019].

As shown in Figure 14, strong intensifications of relativistic and ultra-relativistic electron fluxes have been observed at by
Van Allen Probe B following    an interplanetary shock impact on the Earth's magnetosphere during the 16 July 2017 SSC.
This is the result of ultra-relativistic electrons in the outer radiation belts interacting with the interplanetary shock excited
ULF waves.

The relativistic and ultra-relativistic electron fluxes are oscillating strongly in the ULF Pc5 frequency range (Figure 14).
For a relativistic electron with an energy above ~1 MeV, the oscillation periods modulated by the ULF waves are close to its
drift period in the magnetosphere. Thus, the evolution of energy spectrogram modulated by ULF waves resembles energetic
electron injection with its drift echoes. At lower energy, nevertheless, the electron oscillation period is controlled
predominately by ULF waves, which is almost independent on its energy.

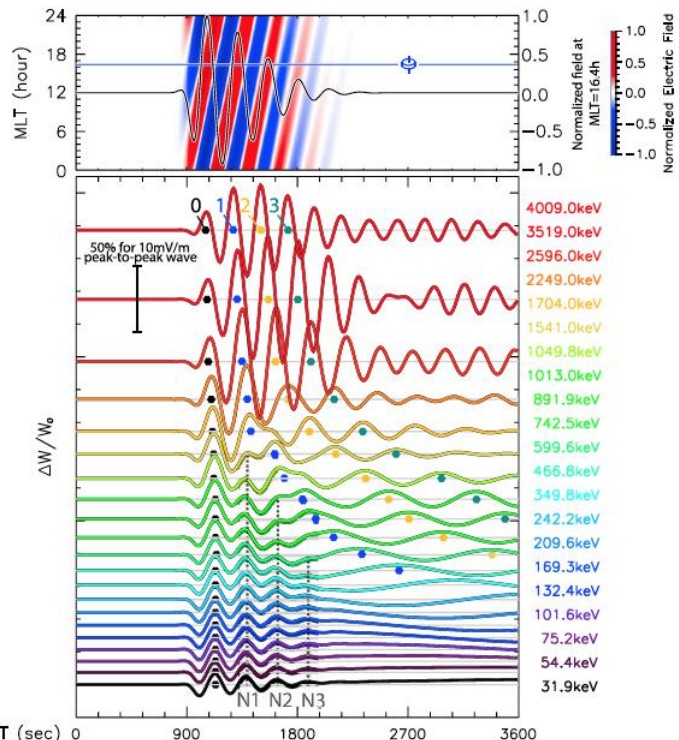

Figure 15: Simulated electron responses in comparison with observations shown in Figure 13. The normalized electric
field is a function of magnetic local time and time. The imaginary probe is placed at MLT = 16.4 hr, as marked with



the blue line. The electric field near the imaginary probe has been overplotted with the black line. (bottom) Predicted

net energy gain of electrons in the electric field of the m = 1 poloidal wave with a sudden onset and a fast-damping

stage in their undisturbed drifting motions. The peaks of energy gain in time sequence of the near-resonant channels

marked with colored dots 0, 1, 2, and 3 correspond to the first (dispersionless) and following (dispersive) stripes 0, 1,

2, and 3 in Figure 13. The vertical dashed lines N1, N2, and N3 refer to the simulated energy dispersionless

modulations in the energy channels far from resonance. MLT = magnetic local time *[Hao et al, 2019]*.


According to the generalized drift resonance theory on charged particles resonating with growth and damping ULF waves [Zhou et al. 2016, Zong et al 2017], the frequency of charged particle flux modulations will shift from the wave frequency to

$m \, \omega_d$ if the ULF waves have decayed, and tilted stripes would be formed in the energy spectrum. When ULF waves disappeared, the formed acceleration and deceleration of charged particle stripes will keep drifting with their respective

speeds. Energy-dependent drift motion along the drift orbit between the interaction region and the spacecraft causes  the charged particle flux oscillation (Figure 15).

As indicated in Figure 15, the spacecraft observations and numerical simulations based on the generalized charged particles drift resonance theory with both growth and damping ULF waves and localized ULF waves agree each other extremely well.





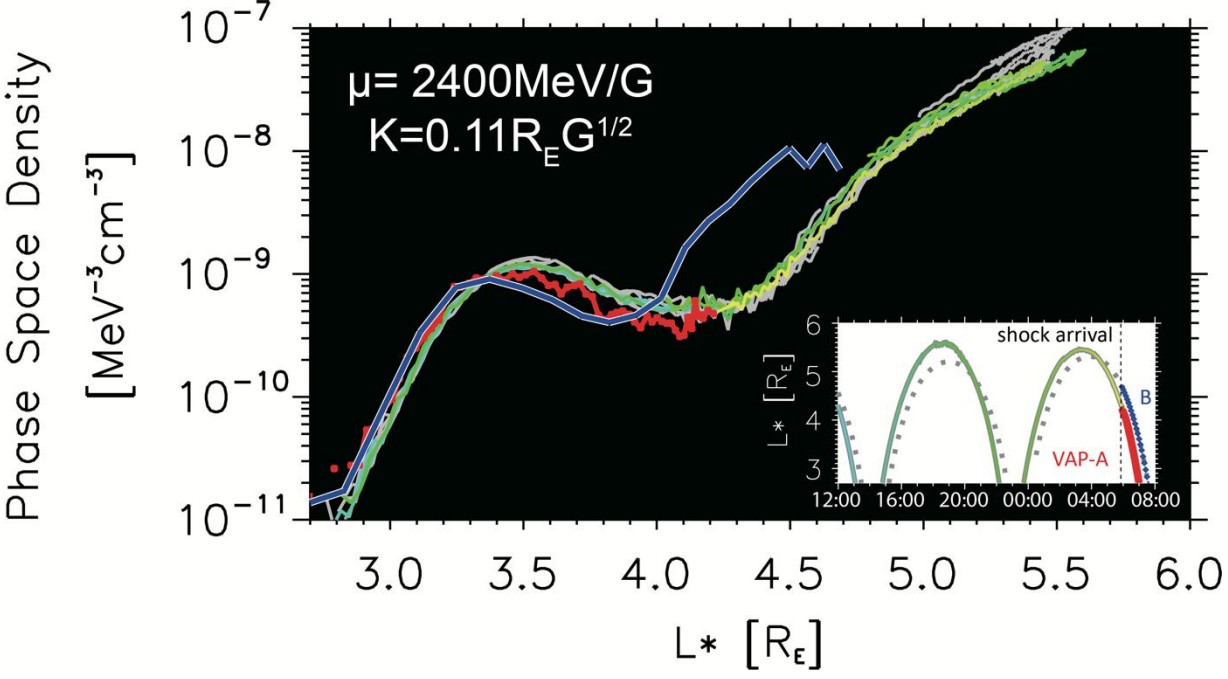

Figure 16. Radial profile of electron phase space density (PSD) at μ=2400MeV/G, K=0.11R_E*G^1/2 measured by REPT telescopes onboard Van Allen Probes. Red (blue) curve presents the PSD measurement from Probe A (Probe B) after the shock arrival. Blue-to-green (gray) curve presents the PSD measured by Probe A (Probe B) before the shock arrival. (Inset) The L* value as a function of universal time of Van Allen Probes from 12:00 July 15 to 8:00 July 16, 2017. Vertical dashed line denotes the shock arrival.

Figure 16 presents the phase space density of ultra-relativistic electrons during the July 16, 2017 interplanetary shock. Before the shock arrival, the PSD distribution $f(L*)|\mu,K$ remained almost unchanged. After the shock arrival, the electron distribution was significantly modified by the interplanetary shock impact within 2 hours and the PSD enhancement of over an order of magnitude is found at 4< L* <4.5 [Hao et al, 2019].

It has been found that the shock induced ULF waves with azimuthal wave number of 1 was the dominant component. Within an hour, the relativistic electron can be accelerated as much as more than 10 times in terms of electron flux (Figure 16) by observed ULF waves [Hao et al, 2019]. Therefore, ULF waves are very powerful to accelerate ultra-relativistic electrons in the radiation belt. The energy spectrum of relativistic electrons has confirmed that ULF waves triggered by the interplanetary shock impact can accelerate outer radiation belt ultra-relativistic electrons up to 3.4 MeV very efficiently in less than an hour (Figure 16). Also, when an interplanetary shock impinging on the magnetosphere, besides the initial





adiabatic acceleration, the spectrum of magnetospheric electrons will be rotated first [Wilken et al., 1986]. Further, additional acceleration can happen via drift resonance with ULF waves [Zong et al, 2009, Zong et al 2017].

In brief, the radiation belt ultra-relativistic electrons can be effectively accelerated by interplanetary shock induced ULF waves within an hour . It has been shown these observed complex and mixed signatures are in consistence with the generalized drift resonance between relativistic electrons and localized ULF waves with both growth and damping features. The observed main features of ultra-relativistic electrons can be reproduced well by numerical results based on the generalized ULF wave – particle drift resonance scenario. This suggests that the generalized drift resonance theory with both growth and damping ULF waves and localized ULF waves is valid and needs to be taken into account for the radiation belt dynamics.

## 4. Generalized theory on the drift-bounce resonance

In this section, the classical drift-bounce resonance concept will be introduced first. Then, a more generalized theory will be described on charged particles' drift - bounce resonance with growth and damping ULF waves. Finally, I will show how poloidal ULF waves interacting with cold plasmaspheric population and the ionospheric outflow.

As mentioned in the above section, the classical drift-bounce resonance condition can be expressed as: $\omega - m \cdot \omega_d = N \cdot \omega_b$ , where N is an integer (normally 0, ±1, ±2), m represents the azimuthal wave number, and $\omega$, $\omega_d$ and $\omega_b$ are the ULF wave frequency, the drift and bounce frequencies of the charged particles in the magnetosphere, respectively. In the Earth's magnetosphere, the bounce frequency of an ion (especially heavy ions, e.g. oxygen ions) is close enough to its drift as well as ULF wave frequencies. Thus, the bounce motion must be considered for charged particles–ULF waves interactions. Charged particles' drift and bounce frequencies ($\omega_d$ and $\omega_b$) are dependent on their kinetic energy, thus, the energy of resonant particles can be decided if the ULF wave frequency is already known.

Since the gradient and curvature drifts of charged particles are in the azimuthal direction in the Earth's magnetosphere, thus, the energies of charged particles can be affected significantly by azimuthal electric field oscillations of poloidal ULF waves. This drift-bounce resonance occurs when particles with a certain energy match the local drift-bounce resonance condition. If the ULF waves are the second harmonic, these resonant charged particles could experience a uniform electric field, as shown in Figure 17. This will lead to fast acceleration of charged particles.

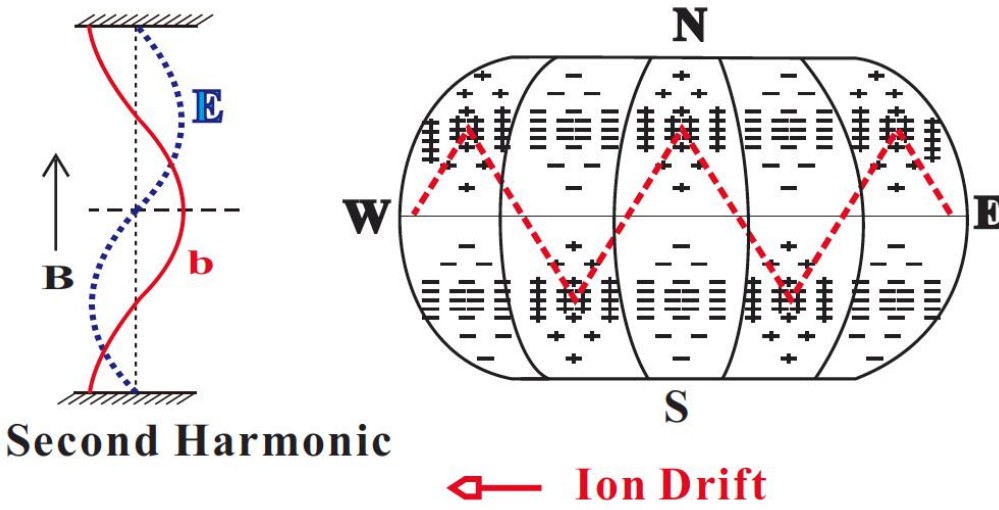

*Figure 17. Fast Acceleration of charged particles by the second harmonic poloidal ULF Waves. A schematic of resonant charged particles satisfying N=1 drift-bounce resonance condition in a second harmonic standing wave. The westward electric fields are indicated by plus and minus, and their magnitudes correspond to the density of the symbols.*


Figure 17, shows the ions satisfying the N=1 drift-bounce resonance condition in a second harmonic ULF wave [Zong et al, 2017]. The particle behaviour is examined in a stretched string model in the wave frame. The westward electric fields are indicated by plus and minus, with their magnitudes corresponding to the density of the symbols. The red dashed lines show the guiding centre orbits of the resonant particles in a second harmonic poloidal ULF wave.

It appears that the resonant ions always stay in the westward wave electric field within each bounce period and will gain a net energy continuously. However, if ions that satisfying the drift-bounce resonance condition in the fundamental mode are considered, these charged particles would experience an accelerating phase (westward electric field) and a decelerating phase (eastward electric field) within a single bounce period and therefore its energy gain can be very small.

Also, if energetic electrons are considered, their guiding centre motion will appear as a vertical line in the second harmonic
ULF waveThe acceleration and deceleration of the electron will cancel out completely over each bounce period. Thus, only in the fundamental mode wave could the electron experience a fast acceleration over a wave cycle as shown in the previous section.





Thus, in principal, energetic ions in the second harmonic poloidal standing waves will be accelerated much more efficiently compared to those in the fundamental mode ULF waves. Furthermore, it has been pointed that the charged particles in the

ring current energy range, e.g., oxygen ions could satisfy all n= ±1, ±2 drift-bounce resonance condition easily. This implies that the drift-bounce resonance is preferred for oxygen ions and is potentially an important mechanism for the ring current oxygen acceleration [Zong et al., 2010, 2012, 2017; Ren et al. 2016, 2017a].

## 4.1    Drift-bounce resonance with growth and damping ULF waves:

"Fishbone" pitch angle distribution

In the classical drift – bounce resonance theory, the ULF wave growth rate is assumed to be time independent, positive, and the amplitude of the ULF wave is extremely small. This is not agreed with satellite observations in the magnetosphere, and the interplanetary shock induced ULF waves are usually with huge amplitude and experience a growth (a positive growth rate) and a damping (a negative growth rate) stage [Tan et al 2004, Zong et al, 2009, Zhang et al 2010, Liu et al

2010]. Thus, a more generalized theory dealing with time dependent ULF waves' interaction with charged particles is required.

The change rate of the particle's kinetic energy within the growth and damping stage of the waves is given by [Zhu et al, 2020, Ren et al. 2019a]:

$$\frac{dW}{dt} = \Sigma_{N=-\infty}^{\infty} \dot{W}_N e^{iN\theta} g(t) e^{i(m\omega_d + m\phi_0 - \omega_r)t} \quad (15)$$

Here conventional notations are used. $g(t)$ describes the growth and damping of the waves:

$$g(t) = \begin{cases} e^{\gamma_1(t-t_0)}, & t < t_0 \\ e^{-\gamma_2(t-t_0)}, & t > t_0 \end{cases} \quad (16)$$

For odd harmonic waves:





$$\delta W \approx \Sigma_{l=0}^{\infty} \dot{a_{2l}} \times (-i) \times \frac{1}{2}\left(\frac{\cos 2l\theta + i\sin 2l\theta}{2l\omega_b + m\omega_d - \omega_r} + \frac{\cos 2l\theta - i\sin 2l\theta}{-2l\omega_b + m\omega_d - \omega_r}\right) e^{i(m\phi - \omega_r t)}$$

$$= \Sigma_{l=0}^{\infty} \dot{a_{2l}} \times \frac{i(\omega - m\omega_d)\cos 2l\theta - (2l\omega_b)\sin 2l\theta}{(m\omega_d - \omega)^2 - (2l\omega_b)^2} \times e^{i(m\phi - \omega_r t)} \tag{17}$$

For even harmonic waves:

$$\delta W \approx \Sigma_{l=0}^{\infty} \dot{b_n} \times \frac{1}{2}\left(\frac{\cos(2l+1)\theta + i\sin(2l+1)\theta}{(2l+1)\omega_b + m\omega_d - \omega_r} - \frac{\cos(2l+1)\theta - i\sin(2l+1)\theta}{-(2l+1)\omega_b + m\omega_d - \omega_r}\right) e^{i(m\phi - \omega_r t)}$$

$$= \Sigma_{l=0}^{\infty} \dot{b_n} \frac{-i(\omega - m\omega_d)\sin(2l+1)\theta - (2l+1)\omega_b \cos(2l+1)\theta}{(\omega - m\omega_d)^2 - (2l+1)^2\omega_b^2} e^{i(m\phi - \omega_r t)} \tag{18}$$

The simulation based on the generalized theory of drift - bounce resonance [Zhu et al. 2020] is employed to reproduce the
time evolution of the pitch angle distributions of energetic protons observed by Van Allen Probe A on 28 January 2014
(Figure 18). This event was first reported by Yamamoto et al. (2019), however, the temporal variations of inclination angles
of each "fishbone" is not addressed.

As illustrated in Figure 18, the inclination of pitch angle stripes increases, "Fishbone-like" structures appear in the electron
pitch angle distribution when the waves are growing [Liu et al., 2020]. According to the generalized drift-bounce resonance
theory [Zhu et al., 2020, Liu et al 2020, Ren et al. 2019a], the increasingly inclined stripes are the manifestation of increasing
phase shift across resonant pitch angles. These observational features can be well predicted by the generalized drift-bounce
resonance theory. The right column of Figure 18 shows the simulation result. A notable feature is the time change of pitch
angles at which flux oscillation is strongest. In other words, the resonant pitch angle changes with time. The black dashed
lines illustrate this tendency. At the beginning, protons resonate at middle pitch angles, e.g. ~ 60° and ~ 120°, whereas at the
end, the resonance pitch angle of hydrogen ions is slightly moving away from middle pitch angles.



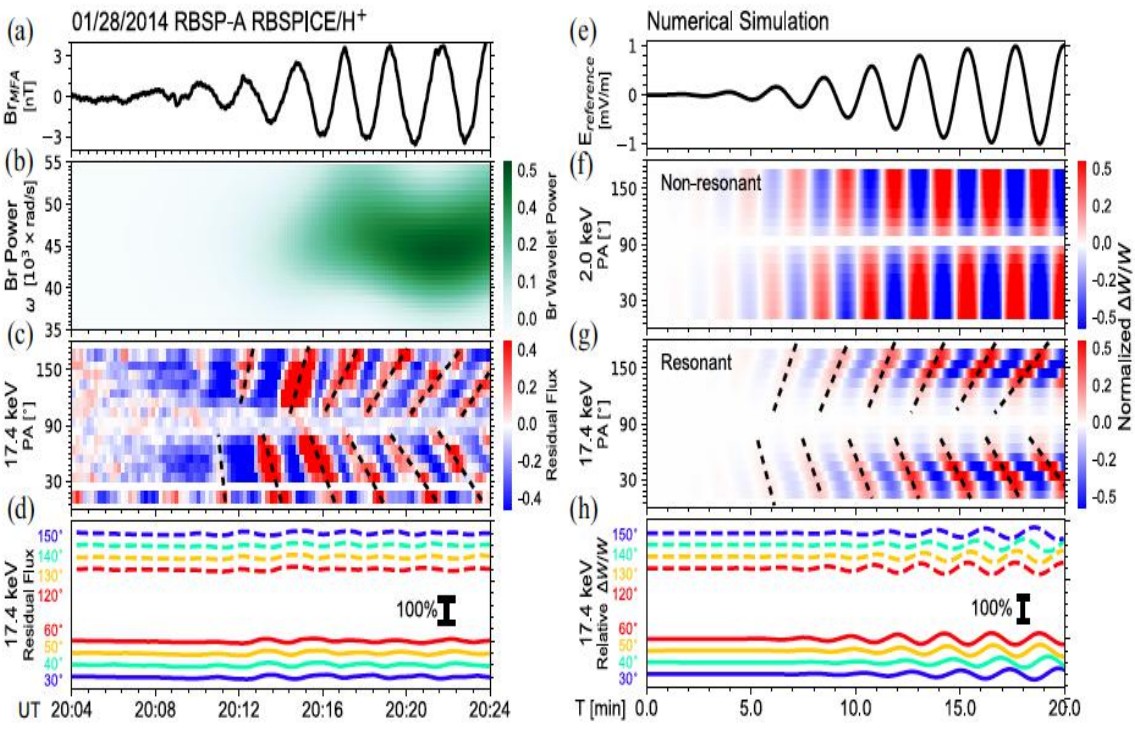

Figure 18. The pitch angle distribution of protons. (a)-(d) Van Allen Probe A observations. During the shown time interval, the waves grow. (a) Radial component (Br) of magnetic field. (b) Dynamic power spectrum of Br. (c) Pitch angle distribution of 17.4 keV protons. The color code shows residual flux. As illustrated by the black dashed lines, the pitch angle stripes become more and more inclined. (d) Line version of (c). (e)-(h) Numerical simulation. (e) Electric field in the simulation observed by a virtual off-equatorial spacecraft. (f) Pitch angle distribution of 2.0 keV protons, which are far away from resonance. The color code shows normalized ΔW/W, which can be directly compared with residual flux. (g) Pitch angle distribution of resonant, 17.4 keV protons. The color code shows normalized ΔW/W [Liu et al, 2020].

Drift-bounce resonance with growth and damping ULF waves can result in the increasingly inclined pitch angle stripes. When the amplitude of the ULF wave is in growing, the stripes of the hydrogen ion pitch angle become more and more inclined. It is shown in Figure 18 that the ULF waves resonate with 17.4 keV hydrogen ions at pitch angles around ~ 40∘ and 140∘.






At the beginning of the wave growth stage, the wave growth rate is large enough to "hide" the phase shift, causing relatively vertical stripes. Then, as the wave grows and its growth rate decreases to zero, the "hidden" phase shift gradually appears, causing the stripes to become more and more inclined. "Fishbone-like" pitch angle structures, thus, is formed by interaction with growth & damping ULF waves [Liu et al, 2020].


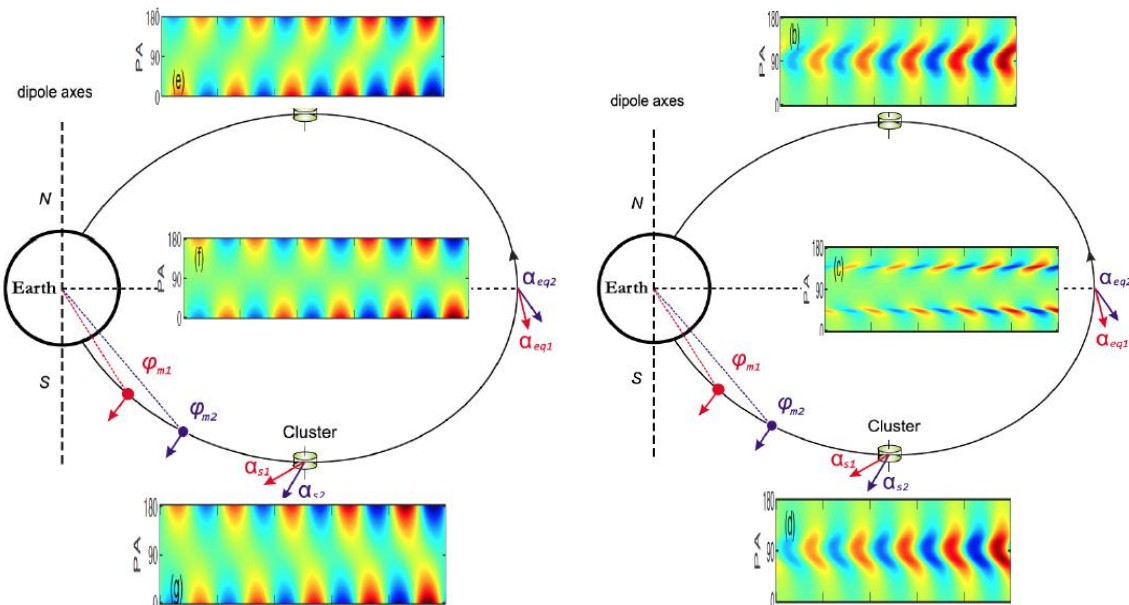

*Figure 19. Generalized drift-bounce theory prediction on resonant ions with the second harmonic ULF waves in the southern hemisphere, magnetic equator, and northern hemisphere. left and Right: Non-resonant and Resonant particles. A*
*schematic of the time of flight effect as ions bounce along a magnetic field line. The red and blue ions have the same energy but different equatorial pitch angles. The symbols $a_{eq}$ and $a_s$ denote the equatorial pitch angle and the local pitch angle detected by the virtual satellite, respectively. Pitch angle distributions at different latitudes are shown qualitatively. [after Zhu et al, 2020].*

Figure 19 summarizes how non-resonant and resonant energetic ions respond to the second harmonic growth and damping ULF waves as observed in the southern hemisphere, magnetic equator, and northern hemisphere. By analysing spacecraft



observations and reproducing them via the generalized drift-bounce resonance theory, it is found that time-varying phase shift across resonant pitch angles can indeed occur, and the effect caused by growth or damping of ULF waves is significant. As a result, the inclination of pitch angle stripes would increase or decrease with time, causing "Fishbone-like" pitch angle

structures.

It is important to note here that "Fishbone-like" structures in ions' pitch angle distribution observed by Van Allan Probes and THEMIS spacecraft can be well reproduced by the generalized drift-bounce resonance theory, therefore provide a more realistic picture of drift-bounce resonance in the Earth's magnetosphere. Therefore, it is important to investigate the

influence of the temporal variations of the wave growth rate on the flux oscillations and their phase shift. The generalized

drift-bounce resonance theory sheds light on the wave-particle interaction between charged particles and ULF waves.

## 4.2 ULF waves' interaction with cold plasmaspheric charged particles

How does plasmaspheric charged particles of a very low energy (~eV) response to ULF waves? For a plasmaspheric

charged particle with energy of a few eV, its drift frequency is much smaller than the bounce frequency: $\omega_d \ll \omega_b$. Therefore, the drift-bounce resonance between the plasmaspheric charged particles and the ULF waves should be dominated by the bounce resonance: $\Omega = N \cdot \omega_b$.

.



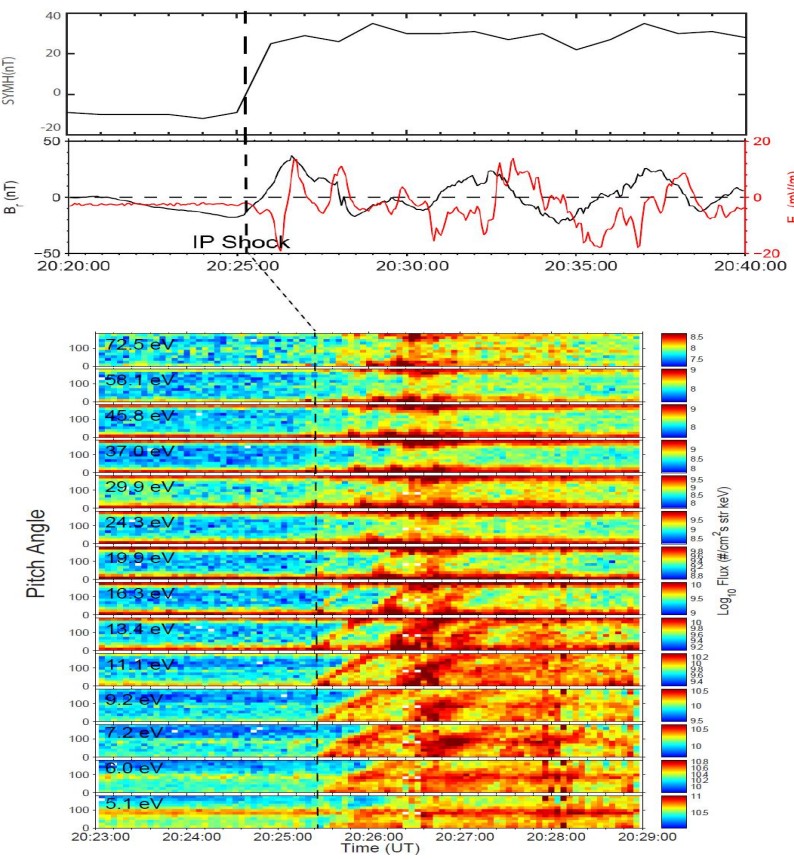

*Figure 20: Top panel show SYM-H index, the poloidal (Br, Ea) wave magnetic and electric fields. And pitch angle distributions of plasmaspheric electrons response to the interplanetary shock impact. Representative electrons with pitch angle distributions for fourteen energy channels ranging between 5 eV and 72.5 eV. The vertical dashed line marks the arrival time of the interplanetary shock at 20:25:10UT. [Zong et al, 2017b]*

However, Once the drift-bounce resonance condition being satisfied, cold plasmaspheric electrons can still be affected by the poloidal mode ULF waves (e.g., Pc5 band). Cold plasmaspheric electrons experience acceleration by the azimuthal electric field of poloidal mode ULF waves which is similar to drift-bounce resonance of oxygen or hydrogen ions [Zong et al, 2017, Ren et al, 2017a].



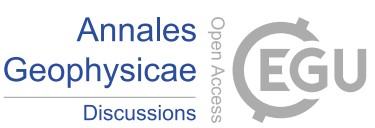

For the plasmaspheric population, the cold electron drift frequency ($\omega_d$) would include both the gradient and curvature drift

term ($\omega_{d.gc}$), the convection E × B drift term ($\omega_{d.Econ×B}$) and the plsmaspheric corotation electric fields term ($\omega_{d.Ecor×B}$)

[Ren et al, 2017b],

$$\omega_d = -\frac{6WLP(\alpha)}{qB_E R_E^2} + \frac{2\Psi_0 L^3 \sin\Phi}{B_E R_E^2} + \Omega_E$$

(19)

where $P(\alpha) = 0.35+0.15 \sin\alpha$ (Hamlin et al., 1961), $\alpha$ is the charged particle equatorial pitch angle, W is the particle energy,

L is the McIlwain L shell value, $B_E$ is the magnitude of Earth's magnetic field at the equator on the Earth's surface, $R_E$ is

Earth's radius, $\Psi_0$ is the electric potential causing the plasma convection in the magnetosphere, $\Phi$ is the azimuthal angle and

$\Omega_E$ is the angular frequency of Earth's rotation.

For a given plasma electron with energy between 1 eV to 1 keV, the drift-bounce resonant conditions for N = 1 and N = 2

can be satisfied with a ULF wave number |m| < 100 [Ren et al, 2017b, 2018, 2019b]. As we can see from *Figure 20*, a sharp

enhancement at SYM-H index has been observed, indicating the interplanetary shock arrival. ULF waves with a large

amplitude oscillation (~15 mV/m) have been observed immediately after the interplanetary shock impinging on the

magnetosphere.

Outstanding and surprising features are that both energy and pitch angle dispersion signatures of plasmaspheric electrons

with an energy of 6 eV to 19.9 eV have been observed clearly. In the dispersion, the electron with a small pitch angle (almost

the field-aligned (0°)) has been observed first, whereas the anti-field-aligned (180°) electrons are observed at last. Different

from the lower energy plasmaspheric electrons, one can see the pitch angle of a higher energy (above 19.9 eV) electron

oscillates between 0° and 180°, and the pitch angle dispersion signature cannot be seen clearly. The period of these





successive dispersion signatures is found to be ~40 s, the same as the observed ULF wave period (third harmonic). Therefore, these multi-dispersions are the results of electron bounce resonance with the interplanetary shock induced ULF waves.

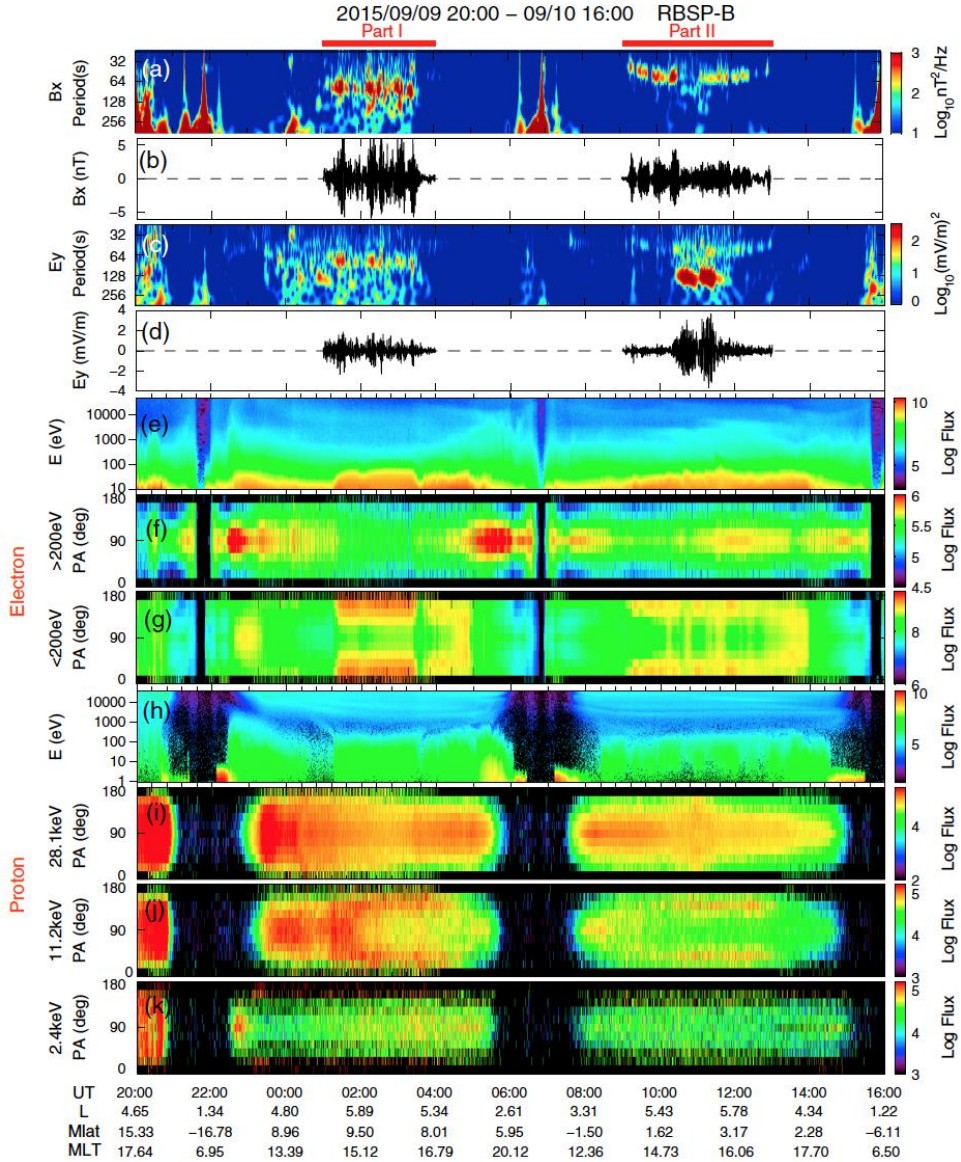

*Figure 21: Overview of Van Allen Probe B observations on ULF wave power spectra, Energy spectra, pitch angle*
*distributions from 20:00 UT on 9 September 2015 to 16:00 UT on 10 September 2015: (a) Wavelet power spectrum of*

*original Bx; (b) Bx component; (c) wavelet power spectrum of original Ey; (d) Ey component; (e–g) electron energy spectrum, pitch angle distributions; (h–k) hydrogen ion energy spectrum, pitch angle distributions [Ren et al, 2017b].*

It is worth pointing out that the ULF wave - particle interaction or plasmaspheric charged particle acceleration region can be determined by backward tracing dispersion signatures of both the energy and the pitch angle. Then, the region of electron acceleration is found to be inside the plasmasphere, it is located off-equator at around –32° in the southern hemisphere.

These can be explained by plasmaspheric electrons interacting with the third harmonic ULF waves with large amplitude electric field at the off-equatorial plasmasphere. The pitch angle dispersion signatures are due to the flux oscillation of "local" non-resonant and resonant plasmaspheric electrons  but not electrons injected from the Earth's ionosphere.

Furthermore, the energy gain of resonant plasmaspheric electrons can be about 20 percent in one wave cycle from the observed interplanetary shock induced large amplitude ULF wave electric field. [Zong et al, 2017]. In general, these results agree with the frame work predicted by the generalized drift-bounce resonance theory (Figure 18).

Figure 21 presents further evidences for both plasmaspheric electrons (<200 eV) and ring current ions (10–20 keV) in
response to ULF waves simultaneously. ULF waves with a period of ~1 minute, which have been observed in two consecutive orbits, lasting several hours. TheULF waves are the second harmonic, thus drift - bounce resonance condition can be satisfied with N=1 for both plasmaspheric electrons and ring current energetic hydrogen ions.

Bidirectional pitch angle distributions for both plasmaspheric electrons and ring current hydrogen ions (10–20 keV) are
observed simultaneously when ULF waves have been observed, and plasmaspheric electron fluxes have been enhanced several times. These observational facts agree with the expectations by the drift-bounce resonance scenario, indicating the importance of ULF waves in the dynamics of plasmaspheric electrons.



## 4.3 ULF waves' interaction with ionospheric outflow: mass spectrometer

Ionospheric outflow is one of the predominant plasma sources of Earth's magnetosphere. It has been shown that the dayside ionospheric outflow ions can interact with ULF waves [Liu et al. 2019, Ren et al. 2015]. It is evident that polarization drift caused by large amplitude electric fields associated with ULF waves may play a significant role in the modulation of singly charged oxygen ions, which may lead to an additional acceleration of oxygen ions [Yue et al., 2016]. This process can be

non-adiabatic if the ULF wave-borne electric field is large enough. It is revealed that the interaction between ULF waves and ionospheric outflow ions occurs predominantly in the perpendicular direction to ambient magnetic field. The cold

ionospheric ions are not only added an energy of $W_{E\times B} = \frac{1}{2} m_i (\frac{|\mathbf{E} \times \mathbf{B}|}{B^2})^2$ by ULF waves to make them "visible" obviously, but also to be separated into ion species according to different mass. The ULF wave modulation on the ionospheric outflow is mass dependent, and this indicates that the ULF wave – charged particle interaction can serve as a mass spectrometer to

distinguish ion species.

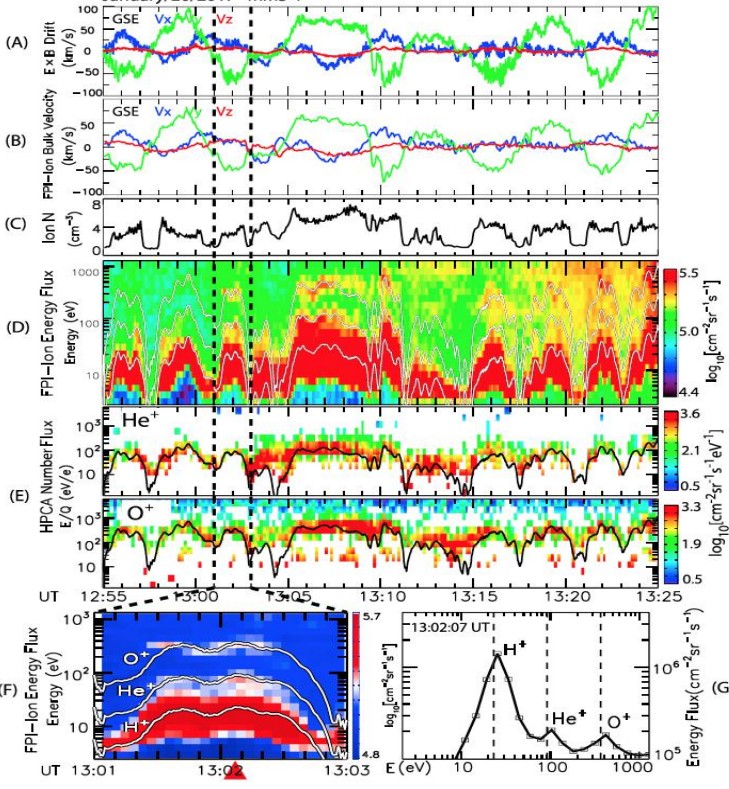




*Figure 22. The responses of cold ions to ultralow frequency waves. (a) E × B drift velocity, derived from magnetic and electric field measurements. (b) Ion bulk velocity. (c) ion number density. (d) The energy-time spectrogram. The three white*
*curves correspond to, from bottom to top, the E × B drift energy of H+, He+, and O+, respectively. (e) The energy-time spectrogram of He+ and O+ number flux. The black curves correspond to E × B drift energy. (f) ion energy spectrogram taken between 13:01 and 13:03 UT. (g)ion energy spectrum at 13:02:07 UT. The white curves in (f) and the black dashed lines in (g) correspond to E × B drift energy [Liu et al, 2019].*

As shown in Figure 22 clearly, ionospheric outflow ions can be modulated by ULF waves driven E x B drift. As a result, the charged particle's energy rises and falls periodically in coincidence to the ULF oscillation. The energy of $H^+$, $He^+$, and $O^+$ ions of ionospheric origin can be added as high as ~75, 300, and 1,200 eV, respectively.

  It is worth pointing out that the effect of polarization drift should be taken into account, due to the large amplitude electric field of the ULF waves. The particle's energy ($W_{total}$) including both E × B drift and polarization drift can be expressed as

$$W_{\text{total}} = \frac{1}{2} m_i (|\frac{\mathbf{E} \times \mathbf{B}}{B^2} + \frac{m_i}{eB^2}\frac{d\mathbf{E}}{dt}|)^2. \tag{20}$$

  The last term equation (20) represents the effect of polarization drift which is proportional to the ion mass. Therefore, polarization drift effect is more profound for heavier ions (oxygen ions) than lighter ions (hydrogen ions).





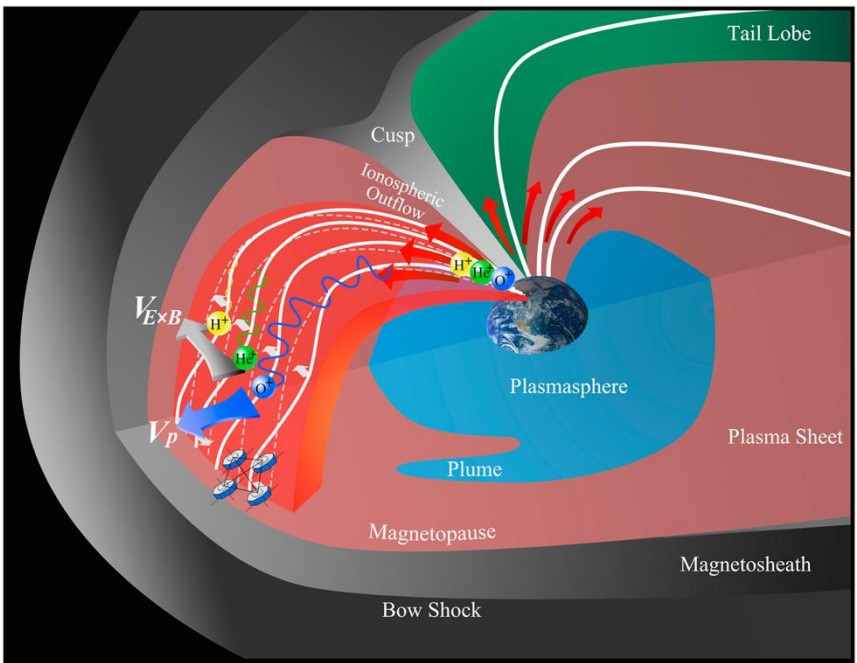


*Figure 23. A schematic of the interaction between ULF waves and ionospheric outflow ions including H+, He+, and O+ (white curves). They are modulated by ULF waves that stand along background magnetic field lines, via ULF wave-induced E × B drift and polarization drift. The E × B drift energy is proportional to ion mass, suggesting that ULF waves can act as a mass spectrometers. Polarization drift also plays a nonnegligible role in the O+ modulation [Liu et al, 2019].*


The observations suggest that the ionospheric heavier ions (oxygen) are modulated significantly by ULF wave-induced $E \times B$ drift and polarization drift (Figure 23). It is shown that the polarization drift is contributed mainly from ULF oscillations whose period is less than 1 minute, whereas only ~20% contributed from oscillations with a period greater than 1 min. It is suggested that $O^+$ can be accelerated significantly by both ULF wave-induced $E \times B$ drift and polarization drift. This

acceleration process is non-adiabatic which agrees with previous theoretical studies (e.g., Cole, 1976; White et al., 2002; Bellan, 2008).

The polarization drift of ionospheric singly charged oxygen ions ($O^+$) induced by ULF wave fields is particularly interesting for magnetospheric physics, since $O^+$ ions can become the dominant ion species (up to 60–80%) in terms of ring current





energy density (Daglis et al. 1999; Zong et al. 2001; Fu et al. 2001; Yue et al., 2019) during magnetic storm time periods. $O^+$ ions in the magnetosphere originate from the Earth's ionosphere. Therefore, it is fundamental important to know how ionospheric singly charged oxygen ions with a few electron volts are accelerated to tens kilo- electron volts and become one of the important magnetospheric components.

## 4.4 Off-equatorial minima effects on ULF wave-particle interaction in the dayside outer magnetosphere

In the inner magnetosphere dominated by the dipole field, the bounce and drift frequencies of charged particles are unimodal functions of pitch angle from 0° to 180° [Hamlin et al., 1961]. However, in the dayside outer magnetosphere, there exists off-equatorial magnetic field minima due to solar wind compression, which can change the trajectories of particles, 795     forcing the orbits of particles with pitch angle near 90° to bifurcate and form the so-called Shabansky orbits [Shabansky, 1971]. Figure 24 shows the trajectory of a Shabansky particle and the magnetic field profiles. Running in an image-dipole magnetic field model, the trajectory of the test particle with pitch angle near 90° bifurcates in the dayside magnetosphere, as shown in panels a-b. Since the magnetic field strength along one field line gets its minima off the equator in the dayside (the red line in panel c), particles with pitch angle near 90° will bounce between two mirror points in the high-latitude minima. 800     Through affecting the bounce and drift motions of particles, off-equatorial minima also modify corresponding frequency – pitch angle relations and change the conventional ULF wave-particle interaction pattern in the inner magnetosphere.



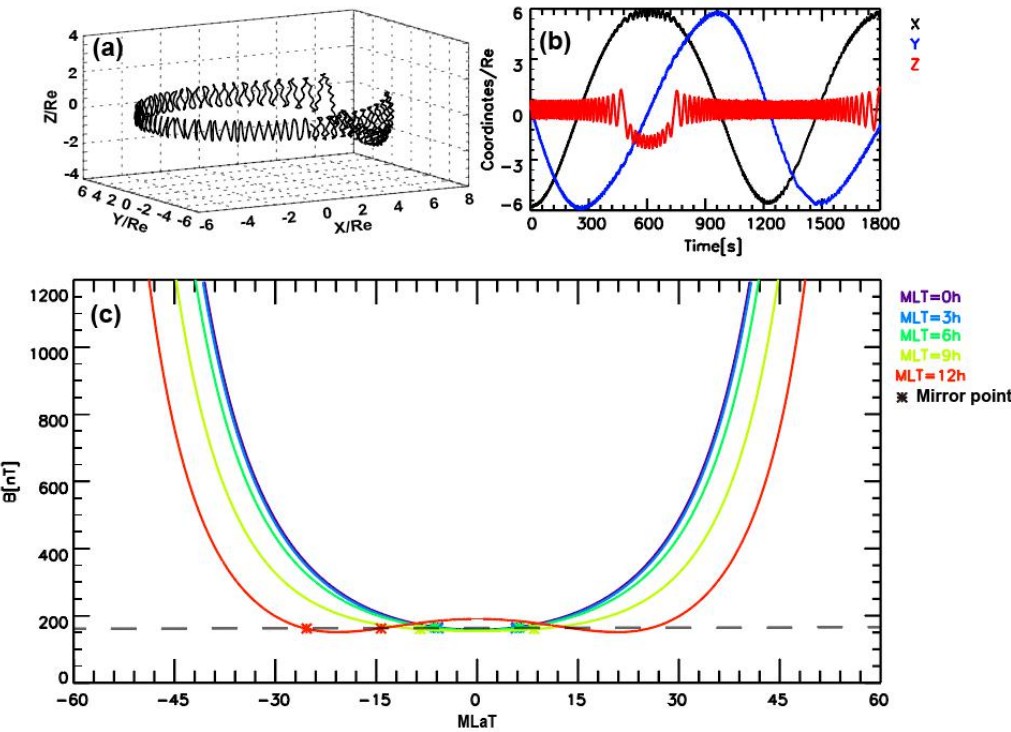

Figure 24. The trajectory projections and magnetic field profiles of a Shabansky particle running in an image-dipole magnetic field. (a-b) The trajectory of a Shabansky particle in GSE coordinates and its time-varying locations. (c) The magnetic field strength on the drifting shell of a Shabansky particle at different MLTs, with asterisk points referring to corresponding mirror points.

Figure 25 shows the pitch angle distributions of a fundamental mode ULF wave-ion interaction event observed by Magnetospheric Multiscale (MMS) on January 20, 2017 [Li et al., 2021]. MMS was located near the subsolar magnetopause for this event. The cold (< 1 keV) ion responses in this event have been studied by Liu et al. [2019], which has been mentioned in Section 4.3. This work focuses on the energetic (> 1 keV) ion responses. The spectrograms show series of quasi-periodical twisted pairs, forming "Pawtrack-like" pitch angle structures. The arrival of 90°-180° pitch angle ions leads that of 0°-90° pitch angle ions, agreeing with the results of Yang et al. [2011]. The conventional pattern of drift-bounce resonance, manifests as two ~180° phase shifts across resonant pitch angles [Zhu et al., 2020]. However, the spectrogram shows more than two ~180° phase shifts, indicating more than two resonant pitch angles for given energy.



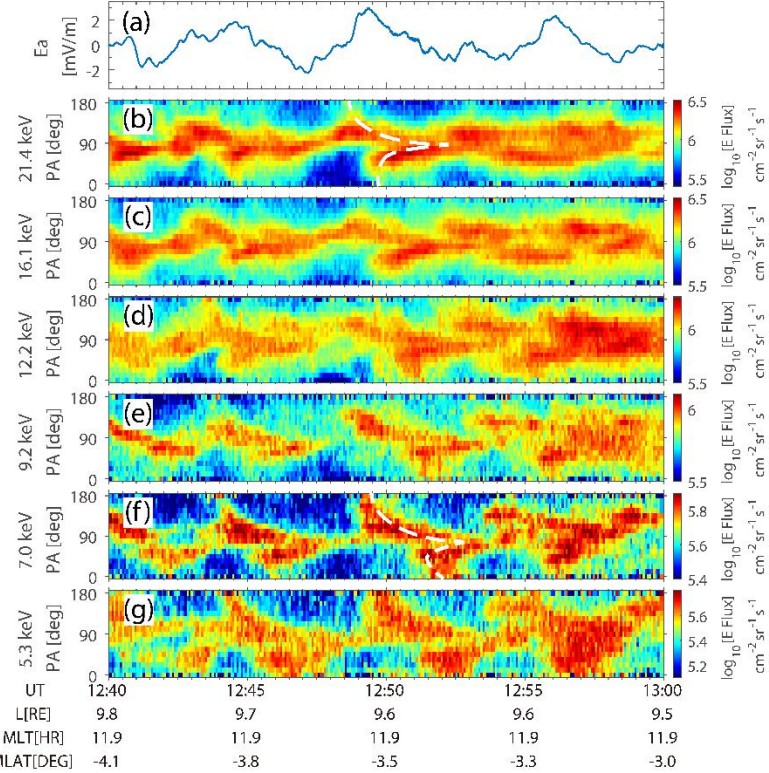

Figure 25. MMS observations on the ULF wave electric field poloidal component and pitch angle distributions of 5.3-21.4 keV ions between 12:40-13:00 UT on January 20, 2017. (a) The azimuthal electric field Ea in MFA coordinates. (b-g) The pitch angle-time spectrograms of 5.3-21.4 keV ions [adopted from Li et al., 2021].

820

Figure 26 illustrates the scenario of how off-equatorial minima affect drift-bounce resonance. Due to the compression of the solar wind, the equatorial magnetic field minimum of the dipole field bifurcates in the dayside outer magnetosphere, forming two off-equatorial minima. The presence of off-equatorial minima changes particles' trajectories and forms two kinds of particles: those with pitch angle close to 90° are trapped into the high-latitude minima and execute Shabansky orbits, while others with larger field-aligned velocity bounce across the equator. Besides, off-equatorial minima modified the frequencies of particle bounce and drift motions. In the inner magnetosphere, the bounce and drift frequencies of particles are unimodal functions of pitch angle. Consequently, there are at most two resonant pitch angles at fixed energy. In the January 20, 2017 event, off-equatorial minima change the bounce (drift) frequency – pitch angle relation from unimodal function to trimodal function (panels c-d in Figure 26), making it possible to form more than two resonant pitch angles at fixed energy. Because of the trimodal shape of the bounce (drift) frequency, every of the 0-75°, 75-105°, and 105°-180° part





of the pitch angle structure corresponds to a group of the conventional drift-bounce resonance pattern, which forms the "Pawtrack-like" pitch angle distribution.

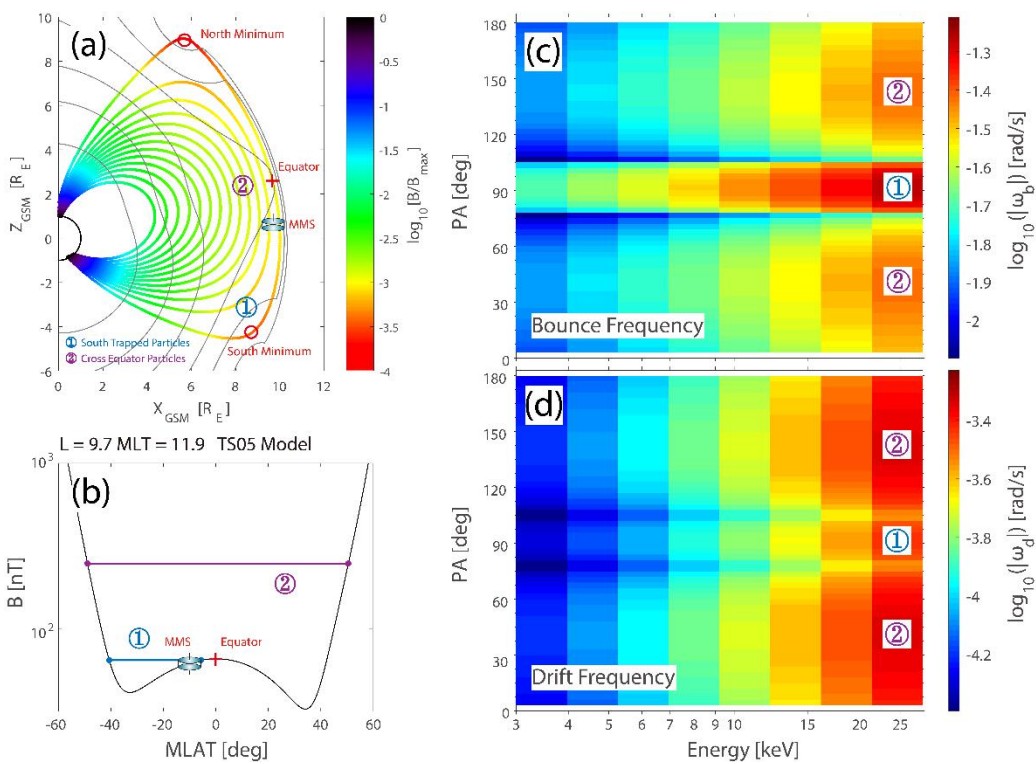

Figure 26. The schematic of off-equatorial magnetic field minima effects on particles' bounce and drift motions. (a) Dayside magnetic field lines and magnetic field strength contours, modeled with the TS05 and the IGRF models. The color codes represent the normalized magnetic field strength $log_{10}[B/B_{max}]$. (b) The modeled magnetic field strength along the magnetic field line where MMS was located (L-shell = 9.7, MLT = 11.9). (c-d) The calculated proton bounce and drift frequencies at the magnetic field line where MMS was located, using the guiding center method described in Roederer and Zhang (2014). The blue ① and purple ② represent particles trapped into the southern high-latitude minima and particles cross the equator, respectively [adopted from Li et al., 2021].

Besides, off-equatorial minima can also affect the trajectories of energetic electrons, leading to abnormal electron drift features on pitch angle-time spectrograms in the dayside magnetosphere. Figure 27 shows the solar wind conditions and pitch angle distributions of energetic electrons observed by the Van Allen Probes on March 11, 2016 [Zhao et al., 2021]. During the time interval 12:51-14:51 UT, both reverse- and normal-boomerang stripes (mentioned in Section 3.2) are observed by two probes, with corresponding solar wind dynamic pressure over 10 nPa. Normal-boomerang stripes indicate



that energetic electrons with 90° pitch angle drift faster at fixed energy, which agrees with the charged particle drift motion pattern in the dipole field [e.g., Hao et al., 2017, Zhao et al., 2020]. On the contrary, reverse-boomerang stripes indicate an abnormal drift velocity – pitch angle relation that particles with 90° pitch angle drift slower, which is opposite to the pattern

of particle drift motion in the dipole field. Test-particle simulations in an image-dipole magnetic field reproduced the observed reverse-boomerang feature at larger L-shells, suggesting that the reverse-boomerang stripes result from off-equatorial minima due to the compression of the magnetopause. In this event, the solar wind dynamic pressure is so large (> 10 nPa) that the off-equatorial minima effects can be observed in the inner magnetosphere (at L-shell ~ 5.9). Meanwhile, normal-boomerang stripes can be observed in the inner region (like L-shell ~ 4.0), where the magnetic field is less affected

by the solar wind dynamic pressure (the magnetic field is expected to be more dipole-like).

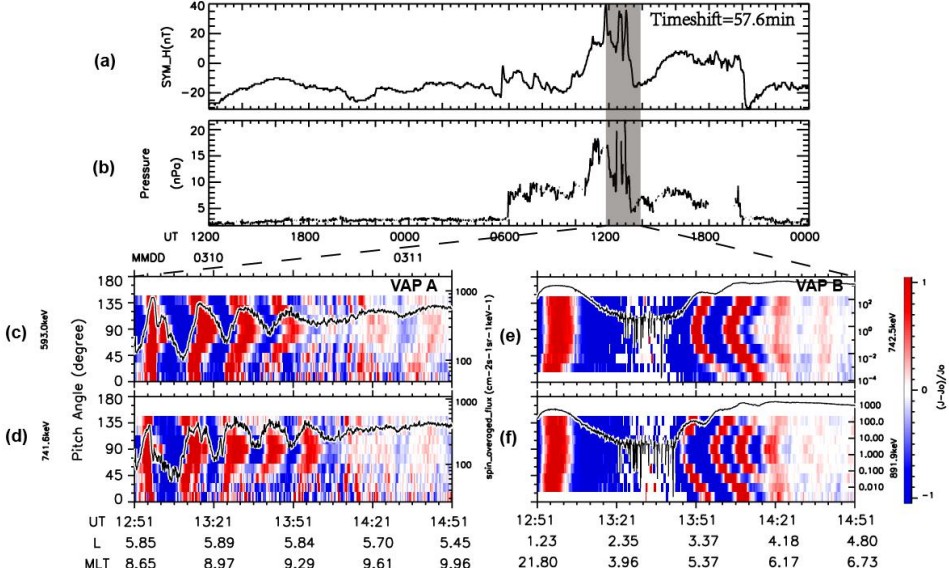

Figure 27. Reverse- and normal-boomerang stripes observed by Van Allen Probes respectively, with the solar wind conditions from OMNI dataset. (a) SYM-H index. (b) The dynamic pressure, with the grey region referring to the corresponding time interval (the time-shift has been considered) of the event below. (c-d) Electron residual flux $(J- J_0)/J_0$, J

and $J_0$ refer to electron origin flux and its 20min running-average) of 593.0 keV, 741.6 keV on pitch angle – time spectrogram. (e-f) Similar with panels c-d but for electron residual flux of 742.5 keV and 891.9 keV measured by MagEIS-B [adopted from Zhao et al., 2021].

However, the electron reverse-boomerang stripes are not as common as the normal-boomerang stripes from the

observations of Van Allen Probes [Zhao et al., 2021], since the orbits of Van Allen Probes are mainly located in the inner magnetosphere. Therefore, reverse-boomerang stripes on electron pitch angle distributions can be observed by Van Allen





Probes only when large compression on the magnetopause forms off-equatorial minima even in the inner magnetosphere. Besides, particles with pitch angle near 90° will bounce between high-latitude mirror points if off-equatorial minima exist in the dayside magnetosphere (panel a in Figure 24). Consequently, localized second harmonic ULF waves could interact with 870 these Shabansky electrons by drift resonance, which have not been reported before.

In conclusion, off-equatorial minima can affect the bounce and drift motions of both ions and electrons, changing the conventional ULF wave-particle interaction pattern. These results reveal new kinds of ULF wave-particle interaction, which potentially affect the efficiency of particle energization for magnetospheric activities relevant to particle energization.


## 5.0    Nonlinear and multiple drift/drift – bounce resonances

In the traditional drift or drift-bounce resonance theory, the weak ULF wave-particle interaction is assumed and charged particle trajectories are unperturbed, thus, a linearization theory can be applied. However, the observed ULF waves in the 880 magnetosphere are usually with a larger magnitude, therefore, the traditional theory needs to be extended into a nonlinear regime since charged particle trajectories are strongly disturbed *[Li et al, 2018, Degeling et al, 2019]*. In this section, the concepts on the nonlinear and multiple drift/drift-bounce resonances will be presented.





## 5.1 Nonlinear drift and drift-bounce resonance

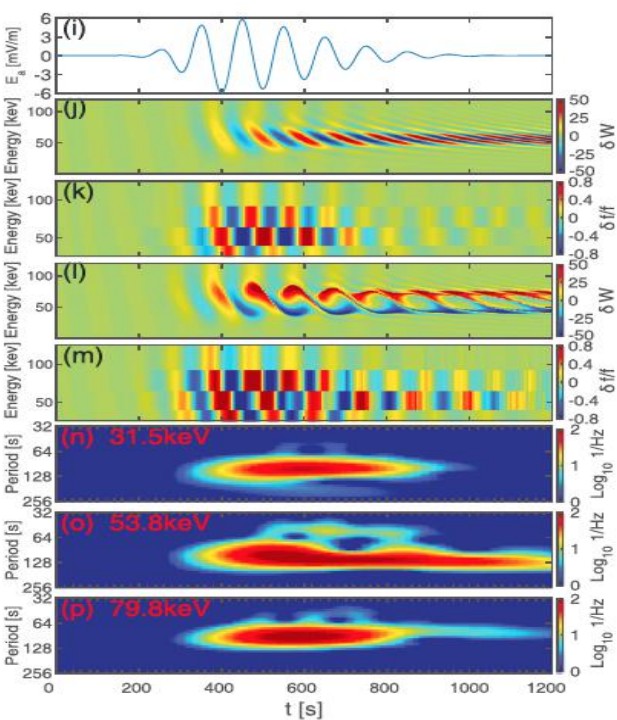


*Figure 28 Comparison of predicted signatures for both linear (tradition theory) and non-linear theory. (i) the panel corresponds to ULF waves with increasing and decreasing amplitudes; (j and l) the energy spectrum of the electron energy gain/loss from ULF waves, obtained from the linear (j) and the nonlinear theories (l); (k and m) the energy spectrum of the electron residual PSD at each energy channel, obtained from the linear (k) and the nonlinear theories (m); (n–p) wavelet*

*power spectrum of the electron residual PSD obtained from the nonlinear theory, in the 31.5-, 53.8-, and 79.8-keV energy channels. [Li et al, 2018]*

A nonlinear theory of drift resonance has been developed to formulate the charged particle motion due to the ULF wave of

a large amplitude *[Li et al, 2018, 2020, Degeling et al, 2019]*. Observable signatures such as rolled-up structures in the

energy spectrum are predicted. As shown in the panel l of Figure 28, the $\delta W$ oscillations are strongest at the resonant energy

of 54 keV, and there appears a sharp, 180° phase shift across the resonant energy. A rolled-up structure eventually appears

at around the resonant energy, this feature could not be predicted by the linear theory.





Such a rolled-up structure has been observed in the energy spectrum by the Van Allen Probes *[Li et al, 2018]*. This provides a solid evidence for the nonlinear drift resonance. The nonlinear drift resonance can be very important in ULF wave

- charged particle interactions in the radiation belts *[Li et al, 2018, Degeling et al, 2019]*.

## 5.2 Multiple drift and/or drift-bounce resonances

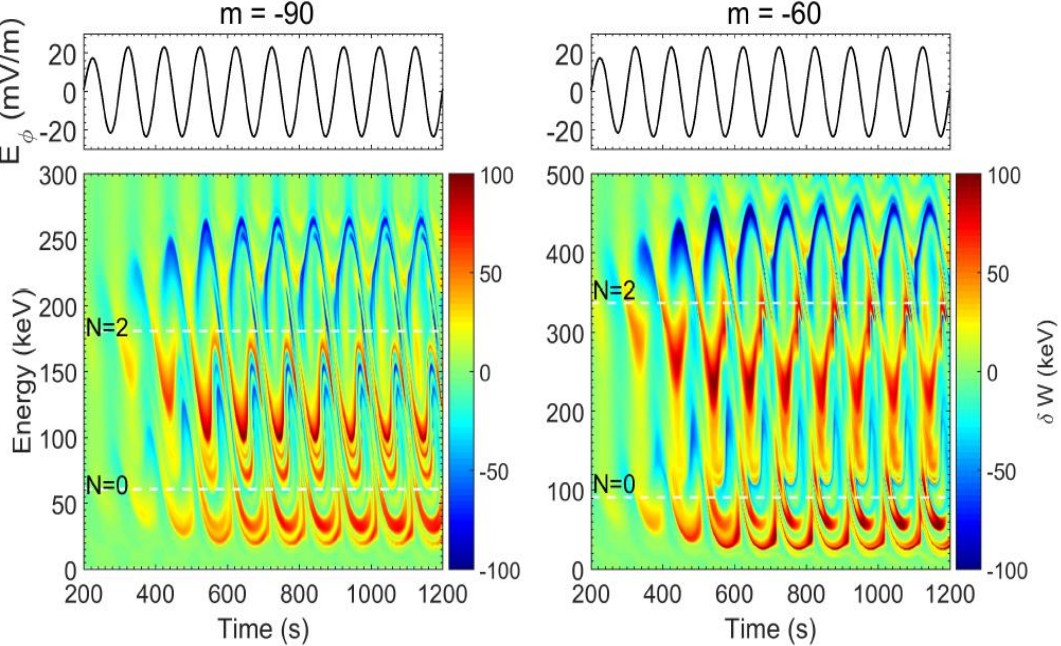

*Figure 29 The interactions between ULF waves and energetic oxygen ions at two different energies. The top-left and top-right panels show the electric field profile as a function of time. The bottom panels show corresponding energy changes, W, experienced by O+ ions as a function of their energy and time on the equator at L=5.7. The wave frequency and azimuthal wavenumber correspond to f ~ 10 mHz m=-90 in the left column, m=-60 in the right column, and the maximum electric field amplitude is 23.8 mV/m at the equator [Rankin et al 2020].*




Multiple drift and/or drift-bounce resonances can occur with different plasma species or the same species at different energies simultaneously. As shown in Fig. 29, it is probable that ULF waves can interact with the energetic oxygen ions at two different energies via both drift-resonance (N=0) and drift-bounce resonance (N=2) simultaneously *[Rankin et al 2020]*.

915 It is found that the oxygen ion differential flux is strongly peaked at the equator. Oxygen flux for drift-bounce resonance peaks at much higher latitudes than those for drift resonance, this can be understood as pitch-angle dependence of the resonance energy.

More observations are needed to verify the features of flux modulations resulting from simultaneous multiple resonances of drift and drift-bounce in more detail, and the resulting ring current dynamics caused by poloidal mode ULF waves in Earth's
920 magnetosphere. Singly charged oxygen ions undergoing drift resonance and drift-bounce resonance can yield new insight into the ring current dynamics of heavy ions that interact with ULF waves.

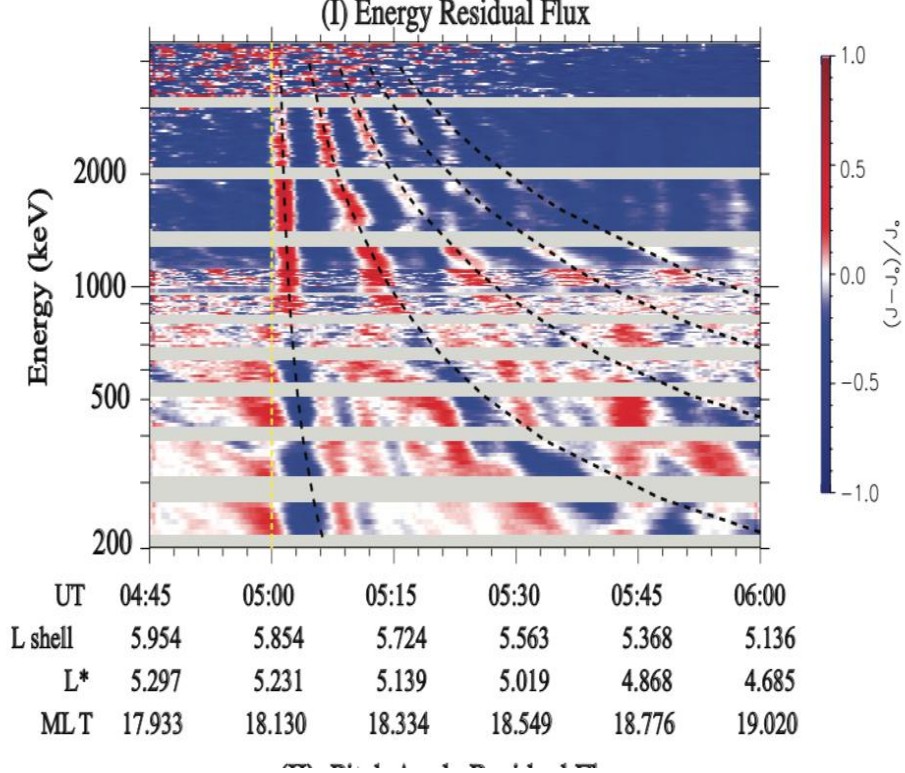

*Figure 30 The residual flux profile of 90° pitch angle electrons of 200-4000 keV based on the high resolution data of MagEIS-B from 04:45 UT to 06:00 UT on May 11th, 2017. The color indicates the value of residual flux. Red means positive, blue means negative and white means nearly zero. The results of estimated electron drift are over-plotted by the black dashed curves. The sudden drop of solar wind dynamic pressure at 05:00 UT is marked by the yellow dashed line. [Ma et al, 2020]*

Another aspect is that multiple ULF waves with different *m* can interact with single plasma population simultaneously. Ultra-high energy resolution data from MagEIS on board Van Allen Probes [Hartinger et al., 2018] in Figure 30 are used to show how magnetospheric charged particles response to a negative solar wind dynamic pressure pulse. As shown in Figure 30, the residual fluxes for electrons with an energy less than 800 keV are decreasing or dropouts, whereas ones with an energy larger than 800 keV are increasing following the negative dynamic pressure pulse arriving. The estimated arrival time of electron drift are over-plotted by the black dashed curves.

The electron fluxes oscillations are consistent with the scenario described in the Section 3.3. For the energetic electron with an energy above ~800 keV, the modulated periods by the ULF waves (low m) are close to its drift periods. However, for energetic electrons with an energy less than ~800 keV, the oscillation is controlled by ULF waves (high m). These mixture signatures are consistent with that energetic electrons at different energies are resonating with ULF waves of different azimuthal wave numbers.

Also, it has been shown that ULF waves can interact with relativistic electrons by drift-resonance and ions by drift-bounce resonance at the same time [Yang et al, 2010, Ren et al. 2016]. Thus, multiple drift and/or drift-bounce resonances can occur simultaneously. These provide a basis for further understanding the dynamic coupling between the radiation belt electrons and the ring current populations in the magnetosphere response to solar wind forcing.

## 6. Outstanding questions and concluding remarks

Magnetospheric physics is now in an extremely vibrant phase with several ongoing and highly-successful missions, e.g. Cluster, THEMIS, Van Allen Probes, and MMS spacecraft, providing  amazing observations and data sets. Since there are

many unsolved fundamental problems, in this paper I have addressed selected topics of ULF wave - charged particle interactions, which encompass many special fields of radiation belt, ring current and plasmaspheric physics. Although great progress has been made over the recent decades, , clear answers have not been found yet as the following:

Do ULF waves mediate coupling between plasmasphere and ring current ion species and radiation belt energetic electrons?
If so, do the ring current ion-excited second-harmonic poloidal ULF waves of moderate m-number cause the energization of radiation belt electrons?

What role do the high-m poloidal mode ULF waves play in the energization of storm-time ring current ions? Is it a prerequisite for a super magnetospheric storm or not?

How common do the high-m poloidal mode waves occur at the plasmapause, and can they be seen as the signature of
existence of the plasmapause? What is the role of the plasmaspheric ion constituency in it? Are high-m poloidal mode ULF waves generated mainly by an exterior solar wind driver or excited by the ring current ions?

What is the role of ULF waves in other planets with a magnetosphere, e.g. Saturn, Jupiter, Mercury's magnetosphere? What is the role of ULF waves in other planets or comets without a magnetic field, e.g. Mars, Venus and Comets?

The response of magnetosphere to the impact by interplanetary shock or solar wind dynamic pressure impulse is not just a
"one-kick" scenario. Instead, the impact generates a series of waves including poloidal mode ULF waves. A generalized theory of drift-bounce resonance with growth or decay and/or localized ULF waves has been developed to explain observations. Energy and pitch angle dependent behaviours for both resonant and non-resonant populations can be well predicted by the generalized drift resonance theory.

The studies on ULF waves' interaction with charged particles will magnificently enrich our understanding of the
interactions of the solar wind and solar wind forcing with the planet's magnetosphere (often cause large geomagnetic disturbances), which is a ubiquitous phenomenon occurring throughout the plasma universe but uniquely accessible within the Earth's magnetosphere. It is realized that the poloidal ULF wave is more effective to accelerate and modulate electrons (fundamental mode) in the radiation belt, as well as charged ions (second harmonic) in the ring current region.

A part of ultra-high energy resolution data already provides us new insight into the drift or drift-bounce resonance,
especially for multiple drift and/or drift-bounce resonances. Any future magnetospheric mission plans should take into



consideration the allowed charged particle detectors to have high energy resolution, high pitch angle resolution and the capability to separate ion mass and charge compositions.

**Data availability.** No data sets were used in this article.


**Competing interests.** The author declares that he has no conflict of interest.

**Acknowledgements.** The study was supported by research grant NSFC Grant Numbers: 41421003, 41627805. I am delighted to acknowledge many collaborators, colleagues-- Xuzhi Zhou, Yongfu Wang, Quanqi Shi, Suiyan Fu, Robert
Rankin, Paul Song, Alex Degeling, as we have shared ideas more than a decade. In particular, I would like to mention Berend Wilken, Theodore Fritz who led me into ESA Cluster mission. My special thanks to Cluster, Double Star, Van Allan Probes, THEMIS and MMS mission for providing the most amazing observations and data sets. The important and fruitful scientific collaborations that I enjoyed are with my talented students Yixin Hao, Ying Liu, Zhiyang Liu, Jie Ren, Xingran Chen, Li Li, Xiaohan Ma and Yifan Zhu of Peking University. Last but not least, I would like to deeply thank my family for
their endless support of my research endeavors.

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
