# Peer review of "Magnetospheric Response to Solar Wind Forcing: ULF Wave - Particle Interaction Perspective"

_Annales Geophysicae, 2021_

## Referee Comment (RC1)

Comments on "Magnetospheric Response to Solar Wind Forcing: ULF Wave – Particle Interaction Perspective" by Qiugang Zong

**General comments:**
As a review paper that was invited as part of the author's EGU Hannes Alfvén Medal, this preprint summaries current understanding and recent advances made by the author's team from Peking University on magnetospheric response to solar wind forcing from the perspective of ULF wave – particle interaction. The solar wind forcing mainly includes interplanetary shock and/or solar wind dynamic pressure pulses. The manuscript is mainly focused on poloidal mode wave interaction with radiation belt energetic electron, ring current ions and plasmaspheric electrons. Generalized theory of drift and drift-bounce resonance with growth or decay localized ULF waves has been discussed and used to explain in situ spacecraft observations. Recent advances on nonlinear and multiple drift/drift-bounce resonance have also been discussed. The manuscript is generally well written and organized with clear presentation, the results described are of interest. I would recommend the paper for publication after my comments below are addressed.

**Specific comments:**
Line 64: "ULF waves containing larger power are the higher frequency ones", in my opinion, it is the opposite way, i.e., ULF waves containing larger power are the *lower* frequency ones.

Lines 79-80: It would be better to mention here that the reason the D component of the ground magnetic field represents the poloidal mode ULF waves is due to the 90-degree rotation when passing through the ionosphere. You mentioned it later in the next paragraph. I would recommend reorganize the two paragraphs so that the logic is clear and the transition is smooth.

Line 180: "The representative values are given in Table 1." You mentioned the *time scales* of the three kinds of particle motions in Equation (2), I am thus expecting the representative values in Table 1 to be time scales instead of frequencies, though they are somewhat interchangeable. If the manuscript is more focused on the wave period and particle motion time scales, I wound recommend listing the time scales in Table 1 as well.

Lines 299 and 303: The Figure caption on line 299 indicates there are 641 IP shock events, but the text on line 303 says 215 shock events. Please double check the reference and be consistent.

Lines 315-320:  Is there a way to separate the contribution of energetic particle acceleration due to the "one-kick" scenario (i.e., bipolar electric fields) and ULF wave-particle interaction? I guess it would be challenging in observations. How about in simulations? Any previous work on this?

Lines 349 and 578: "The eastward (westward) electric fields are indicated by plus and minus". Is it "the azimuthal electric fields are indicated by plus (eastward) and minus (westward)"? Or you mean the waves are eastward (westward) propagating?

Lines 353-354: "since only *uniform* electric field can be experienced by the resonant electrons". I understand what you mean here, but the electric field is not uniform, right? Is there a better word than uniform? The same applies on line 574.

Lines 354-356: Worthwhile mentioning here (or other place), to accelerate electrons in the radiation belt, the wave azimuthal propagation direction should be the same with the electron drift direction, i.e., eastward propagation (positive m-value). The same applies when talking about drift-bounce resonance with ions, a westward propagation wave (or negative m-value) is needed.

Lines 385 and 399: Please describe what $\tau$ is in Equation (6) and $k(\tau)$ in Equation (11).

Line 429: "where m is the ULF wave number, " Please be specific that "m is the **azimuthal** wave number".

Lines 569-570: ULF wave m-value is also needed to determine the resonant energy, right?

Lines 624-625: "Fishbone-like structures appear in the *electron* pitch angle distribution…" Should be **proton** rather than **electron**, right?

Lines 927-931: I cannot find the Ma et al., 2020 and Hartinger et al., 2018 in the reference. Duplicated references also exist, e.g., line 1240 and 1257. Please make sure all the references are listed appropriately.

**Technical corrections:**
Places that need reword:
Line 324-325: "…charged particles resonating with growth and damping ULF waves and charged particles resonating with *azimuthal localized* ULF waves will be described. Finally, I will show how radiation belt relativistic electrons *resonating* with localized growth and damping ULF waves in detail."
Line 786: "Therefore, it is *fundamental important* to know how…"

**Typos/errors:**
Line 160-161: extra (or lack of) space and period.
Line 262: extra parenthesis before the reference.
Line 276: "…suggest that the shock-induced ULF waves *causes* the observed charged particle acceleration"
Line 346: "the bounce motion has no relationship with the ULF *wave* – particle interaction."

Line 402: ", and the phase difference between **difference** electrons with a lower and a higher energy…"

Line 403: "Whereas, **In** the ULF wave damping stage,"

Line 461-462: extra space.

Line 473: "as well **asmodulations**".

Line 480: "**Thus,, it is  crucial** to …" extra comma and space.

Lines 494 and 740: Check the font size of section titles for consistency.

Lines 507-508: "…ultra-relativistic electron fluxes have been observed **at by** Van Allen Probe B following **an   interplanetary** shock…"

Lines 517 and 523: should be Figure 14 instead of Figure 13.

Line 554: extra space.

Line 590: Need a period and space "ULF **waveThe** acceleration…".

Line 731: "**TheULF** waves are…"

Line 952: extra comma.

---

## Referee Comment (RC2)

**General comments:**

This paper aims to summarize our present understanding of the magnetospheric response to solar wind forcing from the ULF wave – particle interaction perspective. Topics addressed include solar wind pressure pulses, poloidal mode waves and their interaction with electrons in the radiation belt, ring current ions and plasmaspheric electrons, focusing on radial transport due to ULF waves. Theoretical, modelling and measurement studies are reviewed.

Summarizing the above topic in a review paper is understandably a formidable task, and it is certainly understood that a lot of important papers will naturally be missed, but several other review papers could be referenced. Such examples are:

— Friedel et al. (2002), Relativistic electron dynamics in the inner magnetosphere – a review, J. Atmos. Solar Terr. Phys., 64(2), 265, doi:10.1016/S1364-6826(01)00088-8

— Shprits et al. (2008), Review of modeling of losses and sources of relativistic electrons in the outer radiation belt I: Radial transport, J. Atmos. Solar Terr. Phys., 70(14), 1679, doi:10.1016/j.jastp.2008.06.008.

— Elkington et al. (2016), The Role of Pc-5 ULF Waves in the Radiation Belts: Current Understanding and Open Questions, in: Waves, Particles, and Storms in Geospace, Oxford University Press, doi:10.1093/acprof:oso/9780198705246.003.0005

A general comment concerns the introductory section: on line 195 the overall organization of the paper is given, including section 1 of the introduction; this could be earlier on, as it reads a bit out of place.

**Minor comments and corrections:**

line 73: "…and are also known as…"

line 109: "Earth's magnetospheric activities" —> perhaps activity in singular form is more appropriate

line 111: "…can take various forms, and most often would excite…"

line 104: "through the ULF wave" —> "through ULF wave"

line 110: the following sentence is repeated in line 112: "The energy coupling between the solar wind and the Earth's magnetosphere can take various forms, most often would excite different plasma waves inside magnetosphere, one of which is the ULF wave."

line 124: "the sudden raise or drop dynamic pressure" —> "the sudden raise or drop of dynamic pressure"

229: "Assumed that a running pulse…" —> "Let us assume that a running pulse…"

237: "is about1 min" —> "is about 1 min"

352: "Once the drift resonance is satisfied" —> "Once the drift resonance condition is satisfied"

403: Whereas, In the ULF wave —> Whereas, in the ULF wave

416: globally

473: as well as modulations

480: Thus, it is crucial

563: how poloidal ULF waves interact with cold plasmaspheric population

731: "TheULF waves" —> "The ULF waves"

---

## Author Comment (AC3)

Comments on "Magnetospheric Response to Solar Wind Forcing: ULF Wave – Particle Interaction Perspective" by Qiugang Zong

General comments:
As a review paper that was invited as part of the author's EGU Hannes Alfvén Medal, this preprint summaries current understanding and recent advances made by the author's team from Peking University on magnetospheric response to solar wind forcing from the perspective of ULF wave – particle interaction. The solar wind forcing mainly includes interplanetary shock and/or solar wind dynamic pressure pulses. The manuscript is mainly focused on poloidal mode wave interaction with radiation belt energetic electron, ring current ions and plasmaspheric electrons. Generalized theory of drift and drift-bounce resonance with growth or decay localized ULF waves has been discussed and used to explain in situ spacecraft observations.
Recent advances on nonlinear and multiple drift/drift-bounce resonance have also been discussed. The manuscript is generally well written and organized with clear presentation, the results described are of interest. I would recommend the paper for publication after my comments below are addressed.

The author really appreciates the referee's valuable comments. We has revised the manuscript accordingly.

Specific comments:
Line 64: "ULF waves containing larger power are the higher frequency ones", in my opinion, it is the opposite way, i.e., ULF waves containing larger power are the lower frequency ones.

*Corrected.*

Lines 79-80: It would be better to mention here that the reason the D component of the ground magnetic field represents the poloidal mode ULF waves is due to the 90-degree rotation when passing through the ionosphere. You mentioned it later in the next paragraph. I would recommend reorganize the two paragraphs so that the logic is clear and the transition is smooth.

*We have rephrased the sentences in the two paragraphs.*

Line 180: "The representative values are given in Table 1." You mentioned the time scales of the three kinds of particle motions in Equation (2), I am thus expecting the representative values in Table 1 to be time scales instead of frequencies, though they are somewhat interchangeable. If the manuscript is more focused on the wave period and particle motion time scales, I wound recommend listing the time scales in Table 1 as well.

We have updated Table 1, adding the periods, and having corrected some numerical value errors.

Lines 299 and 303: The Figure caption on line 299 indicates there are 641 IP shock events, but the text on line 303 says 215 shock events. Please double check the

reference and be consistent.

We have checked the reference. The number of shock events is 215, while the number of dynamic power spectra of the electron fluxes related to these shock events is 641 (There are several geosynchronous observations regarding to one shock event). We have rephrased the caption of Figure 8 to eliminate the confusion.

Lines 315-320: Is there a way to separate the contribution of energetic particle acceleration due to the "one-kick" scenario (i.e., bipolar electric fields) and ULF wave-particle interaction? I guess it would be challenging in observations. How about in simulations? Any previous work on this?

I agree that it's a challenge to quantitatively separate the contribution of energetic particle acceleration due to the initial bipolar electric fields and the following ULF wave-particle interaction using observation only. Studies combining global MHD simulation (to derive the wave magnetic and electric field) and test particle simulation (to evaluate the particle acceleration by the excited ULF waves) with observations are practical to tackle this problem. Mary K. Hudson, XinLin Li and their team have carried out many simulation studies (e.g., ref. 1-3 at the bottom), however to my personal knowledge there's no simulation studies to directly address this question yet. It's surely worth being studied in the coming future.

Lines 349 and 578: "The eastward (westward) electric fields are indicated by plus and minus". Is it "the azimuthal electric fields are indicated by plus (eastward) and minus (westward)"? Or you mean the waves are eastward (westward) propagating?

It indicates the direction of wave electric field, not the direction of wave propagation.

Lines 353-354: "since only uniform electric field can be experienced by the resonant electrons". I understand what you mean here, but the electric field is not uniform, right? Is there a better word than uniform? The same applies on line 574.

"uniform" has been replaced by "one-directional".

Lines 354-356: Worthwhile mentioning here (or other place), to accelerate electrons in the radiation belt, the wave azimuthal propagation direction should be the same with the electron drift direction, i.e., eastward propagation (positive m-value). The same applies when talking about drift-bounce resonance with ions, a westward propagation wave (or negative m-value) is needed.

Agreed. I have added the following sentence: "To meet the resonance condition, the wave azimuthal propagation direction needs to be the same with the particle gradient and curvature drift direction, i.e., eastward propagation wave (positive m) for electrons and westward propagation wave (negative m) for

ions." In line 330-335.

Lines 385 and 399: Please describe what ʌ is in Equation (6) and k(ʌ) in Equation (11).

The descriptions have been added.

Line 429: "where m is the ULF wave number, " Please be specific that "m is the azimuthal wave number".

Corrected.

Lines 569-570: ULF wave m-value is also needed to determine the resonant

energy, right?

Yes. I have rephrased that sentence.

Lines 624-625: "Fishbone-like structures appear in the electron pitch angle

distribution…"
Should be proton rather than electron, right?

Corrected.

Lines 927-931: I cannot find the Ma et al., 2020 and Hartinger et al., 2018 in the reference. Duplicated references also exist, e.g., line 1240 and 1257. Please make sure all the references are listed appropriately.

Corrected.

Technical corrections:
Places that need reword:
Line 324-325: "…charged particles resonating with growth and damping ULF waves and charged particles resonating with azimuthal localized ULF waves will be described. Finally, I will show how radiation belt relativistic electrons resonating with localized growth and damping ULF waves in detail."

Reworded.

Line 786: "Therefore, it is fundamental important to know how…"

Reworded.

Typos/errors:
Line 160-161: extra (or lack of) space and
period. Line 262: extra parenthesis before
the reference.

Line 276: "…suggest that the shock-induced ULF waves causes the observed charged particle
acceleration"
Line 346: "the bounce motion has no relationship with the ULF wave – particle
interaction."
Line 402: ", and the phase difference between difference electrons with a lower and a higher
energy…"
Line 403: "Whereas, In the ULF wave damping stage,"
Line 461-462: extra space.
Line 473: "as well asmodulations".
Line 480: "Thus,, it is crucial to …" extra comma and space.
Lines 494 and 740: Check the font size of section titles for consistency.
Lines 507-508: "…ultra-relativistic electron fluxes have been observed at by Van
Allen Probe B following an interplanetary shock…"
Lines 517 and 523: should be Figure 14 instead of
Figure 13. Line 554: extra space.
Line 590: Need a period and space "ULF waveThe
acceleration…". Line 731: "TheULF waves are…"
Line 952: extra comma.

Corrected.

References:

1. Patel, M., Li, Z., Hudson, M., Claudepierre, S. and Wygant, J., 2019. Simulation of prompt acceleration of radiation belt electrons during the 16 July 2017 storm. *Geophysical Research Letters*, *46*(13), pp.7222-7229.
2. Hudson, M.K., Elkington, S.R., Li, Z., Patel, M., Pham, K., Sorathia, K., Boyd, A., Jaynes, A. and Leali, A., 2021. MHD-Test Particles Simulations of Moderate CME and CIR-Driven Geomagnetic Storms at Solar Minimum. *Space Weather*, *19*(12), p.e2021SW002882.
3. Hudson MK, Elkington SR, Li Z, Patel M. Drift echoes and flux oscillations: A signature of prompt and diffusive changes in the radiation belts. Journal of Atmospheric and Solar-Terrestrial Physics. 2020 Oct 1;207:105332.

---

## Author Comment (AC6)

**General comments:**

This paper aims to summarize our present understanding of the magnetospheric response to solar wind forcing from the ULF wave – particle interaction perspective. Topics addressed include solar wind pressure pulses, poloidal mode waves and their interaction with electrons in the radiation belt, ring current ions and plasmaspheric electrons, focusing on radial transport due to ULF waves. Theoretical, modelling and measurement studies are reviewed.

Summarizing the above topic in a review paper is understandably a formidable task, and it is certainly understood that a lot of important papers will naturally be missed, but several other review papers could be referenced. Such examples are:

— Friedel et al. (2002), Relativistic electron dynamics in the inner magnetosphere – a review, J. Atmos. Solar Terr. Phys., 64(2), 265, doi:10.1016/ S1364-6826(01)00088-8

— Shprits et al. (2008), Review of modeling of losses and sources of relativistic electrons in the outer radiation belt I: Radial transport, J. Atmos. Solar Terr. Phys., 70(14), 1679, doi:10.1016/j.jastp.2008.06.008.

— Elkington et al. (2016), The Role of Pc-5 ULF Waves in the Radiation Belts: Current Understanding and Open Questions, in: Waves, Particles, and Storms in Geospace, Oxford University Press, doi:10.1093/acprof:oso/ 9780198705246.003.0005

*The author really appreciates the referee's valuable comments and remind of these review papers. They have been added in Line 187-188 of revised manuscript.*

A general comment concerns the introductory section: on line 195 the overall organization of the paper is given, including section 1 of the introduction; this could be earlier on, as it reads a bit out of place.

*Agreed. I have moved it to the beginning of the review.*

**Minor comments and corrections:**

line 73: "…and are also known as…"

*Corrected.*

line 109: "Earth's magnetospheric activities" —> perhaps activity in singular form is more appropriate

*Corrected.*

line 111: "…can take various forms, and most often would excite…"

*Corrected.*

line 104: "through the ULF wave" —> "through ULF wave"

*Corrected.*

line 110: the following sentence is repeated in line 112: "The energy coupling between the solar wind and the Earth's magnetosphere can take various forms, most often would excite different plasma waves inside magnetosphere, one of which is the ULF wave."

*Redundancy removed.*

line 124: "the sudden raise or drop dynamic pressure" —> "the sudden raise or drop of dynamic pressure"

*Corrected.*

229: "Assumed that a running pulse…" —> "Let us assume that a running pulse…"

*Corrected.*

237: "is about1 min" —> "is about 1 min"

*Corrected.*

352: "Once the drift resonance is satisfied" —> "Once the drift resonance condition is satisfied"

*Corrected.*

403: Whereas, In the ULF wave —> Whereas, in the ULF wave

*Corrected.*

416: globally

*Corrected.*

473: as well as modulations

*Corrected.*

480: Thus, it is crucial

*Corrected.*

563: how poloidal ULF waves interact with cold plasmaspheric population

*Corrected.*

731: "TheULF waves" —> "The ULF waves"

*Corrected.*